# DIRECT DOUBLY ROBUST ESTIMATION OF CONDITIONAL QUANTILE CONTRASTS

**Josh Givens & Song Liu**
School of Mathematics
University of Bristol
`joshgivens@hotmail.co.uk`

**Henry W. J. Reeve**
School of Artificial Intelligence
Nanjing University

**Katarzyna Reluga**
School of Business and Economics
Humboldt University of Berlin

## ABSTRACT

Within heterogeneous treatment effect (HTE) analysis, various estimands have been proposed to capture the effect of a treatment conditional on covariates. Recently, the *conditional quantile comparator* (CQC) has emerged as a promising estimand, offering quantile-level summaries akin to the conditional quantile treatment effect (CQTE) while preserving some interpretability of the conditional average treatment effect (CATE). It achieves this by summarising the treated response conditional on both the covariates and the untreated response. Despite these desirable properties, the CQC's current estimation is limited by the need to first estimate the difference in conditional cumulative distribution functions and then invert it. This inversion obscures the CQC estimate, hampering our ability to both model and interpret it. To address this, we propose the first direct estimator of the CQC, allowing for explicit modelling and parameterisation. This explicit parameterisation enables better interpretation of our estimate while also providing a means to constrain and inform the model. We show, both theoretically and empirically, that our estimation error depends directly on the complexity of the CQC itself, improving upon the existing estimation procedure. Furthermore, it retains the desirable double robustness property with respect to nuisance parameter estimation. We further show our method to outperform existing procedures in estimation accuracy across multiple data scenarios while varying sample size and nuisance error. Finally, we apply it to real-world data from an employment scheme, uncovering a reduced range of potential earnings improvement as participant age increases.

## 1 INTRODUCTION

As data becomes more and more readily available, the demand for personalised treatments and interventions has increased dramatically. The statistical field addressing this challenge is heterogeneous treatment effect (HTE) analysis in which one aims to learn the effect of a treatment on an outcome or response conditional on key covariates (Hirano and Porter, 2009; Collins and Varmus, 2015; Obermeyer and Emanuel, 2016; Lei and Candès, 2021).

A core strategy for the analysis of HTE data is to estimate key estimands that quantify the effectiveness of a treatment given the covariates. The two commonly used estimands are the conditional average treatment effect (CATE) (Abadie and Imbens, 2002; Imbens, 2004; Semenova and Chernozhukov, 2021) and the conditional quantile treatment effect (CQTE) (Abadie et al., 2002; Autor et al., 2017; Powell, 2020) which represent the difference in the conditional mean and quantile of the response respectively for the two treatments given the covariates. Both approaches have advantages: the CQTE yields more granular treatment-effect summaries and is less sensitive to extreme values (Firpo, 2007; Bitler et al., 2006), while the CATE provides a more interpretable estimand with stronger estimation guarantees (Kennedy et al., 2023; Kennedy, 2023b; Nie and Wager, 2020).

A recently introduced estimand, the conditional quantile comparator (CQC) (Givens et al., 2024), aims to bridge the gap between the CATE and the CQTE. The CQC does this by providing a transport

map between the conditional treated and untreated response distributions in a quantile preserving manner. The definition of the CQC more naturally aligns with how treatment effects are discussed as they are often talked about as either improving the response by a fixed amount or scaling the response (e.g. a medicine increased life expectancy by 2 years or by 50%). This scaling can be expressed naturally as a function of response while we would need to transform the input via the conditional cumulative distribution function of the untreated response in order to express it as a function of the associated quantile. Therefore in this case the CQC would be able to directly capture this effect helping better understand the treatment and its efficacy while the CATE and CQTE would likely have much more complex relationship for the CATE and CQTE. As the CQC shares properties with the CQTE it also shares its strengths. Namely it is useful in settings where our distribution is heavily skewed, such as platform use or income, as it allows us to make effective decision on which treatment is better without being heavily affected by a small number of extreme samples (Firpo, 2007; Belloni et al., 2017). In relation to this, it can also help with decision making in cases where we want to evaluate our treatment only for certain response values. For example if we want to evaluate some employment intervention on income for those on lower incomes (see our example in Section 5.)

In summary, this leads to the CQC being able to give information on the relationships between the treated and untreated distributions at all levels similarly to the CQTE, while having a more direct interpretation at the response level. The current estimation method for the CQC, introduced in Givens et al. (2024), involves estimating an intermediate estimand and then inverting this to obtain a CQC estimate. Despite having some strong theoretical guarantees, this framework does not enable direct modelling of the CQC itself. This in turn prevents the use of informative parameterisations and limits our ability to constrain or inform the model structure, such as by enforcing smoothness in nonparametric settings. This approach also hinders interpretability of the estimate as it can only be examined via evaluating it at various samples, a procedure which itself can be computationally costly.

In this paper, we provide the first direct estimator of the CQC which addresses these limitations. Crucially, our new approach allows the CQC to be explicitly parameterised. This enables us to enforce assumptions on the CQC via flexible techniques including linear parameterisation, neural networks, kernel bandwidth choice in nonparametric settings, and regularisation. This also enhances interpretability by allowing greater flexibility in model inspection. Finally, because our approach models the CQC directly, the estimation error depends on the complexity of the CQC itself, rather than that of an upstream intermediate function. Meanwhile, it retains the doubly robust property, ensuring accurate estimation of the CQC even when all nuisance parameters are estimated suboptimally.

To summarise, in this paper we:

- Provide the first direct CQC estimation procedure.

- Provide finite sample bounds on this estimation procedure.

- Illustrate the robustness of our estimator theoretically, and through numerical experiments.

- Show it to empirically outperform existing procedures in terms of estimation accuracy along various axes directly highlighting the advantage given by our explicit parameterisation.

- Illustrate its interpretability by applying it to real world problems and analysing the results.

## 2 PROBLEM FORMULATION

We first introduce the general HTE setting. Let $Y, X, A$ be random variables each representing information about an individual in our treatment setting. Specifically we take $Y$ (on $\mathcal{Y} \subseteq \mathbb{R}$) to give their univariate outcome/response; $X$ (on $\mathcal{X} \subseteq \mathbb{R}^d$) to give their covariates of interest e.g. age, height, etc.; and $A$ (on $\{0, 1\}$) to give their treatment assignment with $1 =$ Treatment and $0 =$ Control. Our overall aim is to understand the effect of treatment, $A$, on the response, $Y$, given the covariates, $X$.

We define $Z = (Y, X, A)$ and let $D := \left\{ Z^{(i)} \right\}_{i=1}^{2n} \equiv \{(Y^{(i)}, X^{(i)}, A^{(i)})\}_{i=1}^{2n}$ for $n \in \mathbb{N}$ denote IID copies of $Z$ representing our data sample with $i$ indexing each sample/individual and $2n$ used for notational convenience. We assume that we are in the potential outcome framework so there exists $Y(1), Y(0)$ representing an individuals response both on and off treatment such that $Y \equiv Y(A)$. To allow our results to translate back to these potential outcomes we make the no unobserved confounding assumption given by the identity $(Y(0), Y(1)) \perp A|X$. Crucially this means that $Y(a)|X = \boldsymbol{x}$ and $Y|X = \boldsymbol{x}, A = a$ are identically distributed for $\boldsymbol{x} \in \mathcal{X}, a \in \{0, 1\}$ (Rubin, 2005).

For $n \in \mathbb{N}$, let $[n] := \{1, \ldots, n\}$. For a vector $\boldsymbol{w} \in \mathbb{R}^p$ let $w_j$ to represent the $j^{\text{th}}$ component of $\boldsymbol{w}$ and let $\|\boldsymbol{w}\|$ be the Euclidean norm of $\boldsymbol{w}$ unless otherwise specified. For a function $f : \mathbb{R} \times \mathcal{X} \to \mathbb{R}$, we let $\partial_y f(y, \boldsymbol{x})$ denote the partial derivative $\frac{\partial}{\partial y} f(y, \boldsymbol{x})$. Finally, as convention, for $a < b$ we take $\int_b^a f(x) \mathrm{d}x = -\int_a^b f(x) \mathrm{d}x = -\int_{[a,b]} f(x) \mathrm{d}x$. With this notation and basic treatment effect set-up introduced, we can now define key estimands used in our framework.

**Remark 1.** *For simplicity, we will assume that response, $Y$, is continuous with strictly positive density when conditioned upon any covariate, $X$, and treatment, $A$.*

## 2.1 NUISANCE PARAMETERS AND KEY ESTIMANDS

We first define various *nuisance parameters*, which are additional distributional objects necessary for the estimation of our estimand. The three nuisance parameters of interest are the propensity score, $\pi : \mathcal{X} \to (0, 1)$, *conditional cumulative distribution function* (CCDF) of $Y|X, A$, $F_a$, and the *conditional quantile function* of $Y|X, A$, $F_a^{-1}$, each defined as

$$\pi(\boldsymbol{x}) := \mathbb{P}(A = 1|X = \boldsymbol{x}) \tag{1}$$

$$F_a(y|\boldsymbol{x}) := \mathbb{P}(Y \leq y|X = \boldsymbol{x}, A = a), \tag{2}$$

$$F_a^{-1}(\alpha|\boldsymbol{x}) := \inf\{y \in \mathbb{R}|F_a(y|\boldsymbol{x}) \geq \alpha\}. \tag{3}$$

for all $\boldsymbol{x} \in \mathcal{X}$ and $a \in \{0, 1\}$ and with $\pi$ assumed to be continuous and bounded away from $\{0,1\}$. The propensity score can be thought of as the probability of an individual being assigned to treatment given their covariates. Finally we take $p_a(.|\boldsymbol{x})$ to represent the probability density function (pdf) of $Y|X = \boldsymbol{x}, A = a$. We can now introduce the core HTE estimands.

**Definition 1** (CATE, CQTE, CQC)**.** *The CATE, CQTE and the CQC of the triple $Z = (Y, X, A)$ are given by $\tau : \mathcal{X} \to \mathbb{R}$, $\tau_q : [0, 1] \times \mathcal{X} \to \mathbb{R}$, and $g^* : \mathcal{Y} \times \mathcal{X} \to \mathcal{Y}$ respectively with*

$$\tau(\boldsymbol{x}) := \mathbb{E}[Y|X = \boldsymbol{x}, A = 1] - \mathbb{E}[Y|X = \boldsymbol{x}, A = 0],$$

$$\tau_q(\alpha|\boldsymbol{x}) := F_1^{-1}(\alpha|\boldsymbol{x}) - F_0^{-1}(\alpha|\boldsymbol{x}),$$

$$g^*(y_0|\boldsymbol{x}) := F_1^{-1}\left\{F_0(y_0|\boldsymbol{x})|\boldsymbol{x}\right\}.$$

Both the CATE and the CQTE aim to summarise the effect of the treatment by examining the difference in the outcome for the treated and untreated patients given specific covariate values. The CQTE offers added granularity by allowing the effect to be examined at specific quantiles rather than providing a single summary statistic per covariate value.

The CQC is the central focus of our work and differs from previous estimands by instead mapping an untreated response and covariate value to a treated response value (Givens et al., 2024). Specifically, it defines a transport map from the distributions of the untreated response to the equivalent quantile value of the treated response via conditional on the covariates. Previous work has demonstrated the CQC's ability to provide granular quantile level summaries of the treatment effect similarly to the CQTE while framing the input more naturally in terms of an untreated response value as opposed to a quantile level. The CQC achieves this by providing summaries over multiple quantiles similarly to the CQTE and in fact has the relation that $\tau_q\{F_0(y_0|\boldsymbol{x})|\boldsymbol{x}\} = g(y_0|\boldsymbol{x}) - y_0$.

A key strength of the CQC working specifically in the response space, is that this more naturally mimics how the impact of a treatment or intervention is often characterised. Specifically the effect of a treatment is often expressed in terms of either the absolute effect (additive effect) or a scaling effect on the response itself (multiplicative effect.) If this impact is deterministic, the CQC will be able to represent these effects in a simple manner either of these effects while the CQTE may not. Figure 1 provides an example of this when the treatment doubles the response. We plot the CATE, CQTE and CQC and show that both the CATE and the CQTE contain complex high frequency changes not present in this treatment effect while the CQC does not. Specifically, the CQC will be $g^*(y_0|\boldsymbol{x}) = 2y_0$ regardless of the marginal distributions. This relative simplicity of the CQC not only improves interpretability but can also lead to more accurate estimation.

Optimal estimation of the estimands in Definition 1 has been the focus of much previous work (Robins et al., 2008; Shalit et al., 2017; Foster and Syrgkanis, 2023; Melnychuk et al., 2025; Sun and Xia, 2025). To achieve their optimal estimation, it is first necessary to estimate nuisance parameters, such as the propensity score and conditional cumulative distribution functions (CCDFs) in case

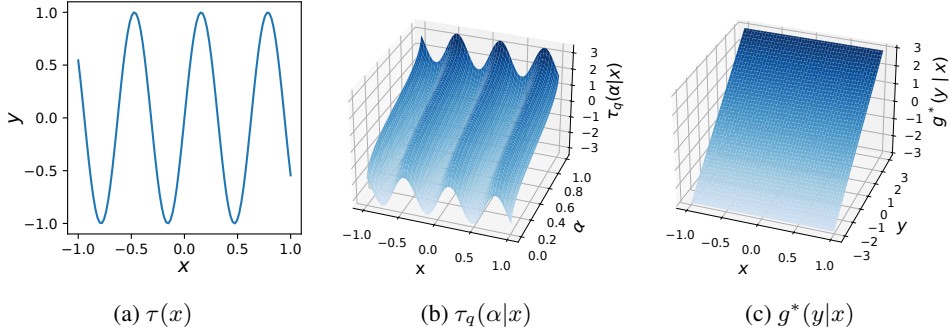

(a) $\tau(x)$       (b) $\tau_q(\alpha|x)$       (c) $g^*(y|x)$

Figure 1: Surface plots for CATE (panel (a)), CQTE (panel (b)), and CQC (panel (c)) where $Y|X = x, A = 0 \sim N(\sin(10x), 1)$, $Y|X, A = 1 \sim N(2\sin(10x), 4)$. We can see that CATE, and CQTE have high-frequency changes in $x$ while the CQC does not depend on $x$ instead simply representing the doubling of the response as $g^*(y|x) = 2y$.

of CQC estimation. Consequently, prior work has focused on developing methods that are robust to inaccuracies in these nuisance estimates. A notable class of such methods, known as *doubly robust* methods, can attain the desired overall convergence rate even when all nuisance parameter estimates converge at slower rates. Doubly robust methods have been introduced for each of the CATE (Kennedy et al., 2023; Kennedy, 2023b; Nie and Wager, 2020), CQTE (Kallus and Oprescu, 2023), and CQC (Givens et al., 2024). We now introduce the existing doubly robust CQC estimation method, which serves as a point of comparison for our proposed approach.

## 2.2 CURRENT CQC ESTIMATION

In Givens et al. (2024) a CQC estimation method was introduced which involved estimating an intermediary function called the *CCDF contrasting function* defined as

$$h(y_1, y_0, \boldsymbol{x}) = F_1(y_1|\boldsymbol{x}) - F_0(y_0|\boldsymbol{x}).$$

To obtain an estimate of $g^*(y_0|\boldsymbol{x})$ one would then have to estimate $h(y_1, y_0, \boldsymbol{x})$ over a large number of $y_1$ samples, isotonically project, and then choose the $y_1$ sample which gave $h$ closest to 0. This approach has three main shortcomings:

1. Its lack of explicit form for our CQC estimate, $\hat{g}$, makes it harder to interpret and constrain.

2. Its estimation quality depends upon the difficulty of estimating $h$ rather than our parameter of interest, $g^*$.

3. Its evaluation is computationally expensive especially when the estimate of $h$ is expensive to evaluate (see Appendix D.3.3 for further exploration and experimental validation of this.

**Remark 2.** *We view simplicity of the CQC as a more natural and easily satisfied notion than that of the CCDF contrast function. See Appendix C.1 for further discussion and an illustrative example.*

We now introduce our approach which directly estimates the CQC, thereby addressing these issues.

## 3 THE DIRECT CQC ESTIMATOR

Similarly to the existing approach, we can frame our estimation problem as finding $y_1$ for a given $y_0, \boldsymbol{x}$ such that $h(y_1, y_0, \boldsymbol{x}) = F_1(y_1|\boldsymbol{x}) - F_0(y_0|\boldsymbol{x}) = 0$. While we could treat this as a Z-estimation problem, in order to extend this to learning a function over all $y_0, \boldsymbol{x}$, it is instead helpful to view it through this lens of M-estimation. To this end, since $h$ is an increasing function of $y_1$, any loss function $\bar{\ell}$ satisfying $\partial_{y_1}\bar{\ell}(y_1, y_0, \boldsymbol{x}) = h(y_1, y_0, \boldsymbol{x})$ will be minimised at the value of $y_1$ such that $h(y_1, y_0, \boldsymbol{x}) = 0$, our desired goal. Using this idea we now introduce our loss in Definition 2, justify it via Equation (4), and demonstrate its direct relation to CQC estimation error in Proposition 1.

**Definition 2.** *For a parameter space $\Theta \subset \mathbb{R}^p$, let $\mathcal{G}_{\boldsymbol{\theta}} := \{g_{\boldsymbol{\theta}} : \mathcal{Y} \times \mathcal{X} \to \mathcal{Y} | \boldsymbol{\theta} \in \Theta\}$ be the set of parameterised CQC estimates. Additionally, for $y_0 \in \mathcal{Y}$, $\boldsymbol{x} \in \mathcal{X}$, $\boldsymbol{\theta} \in \Theta$, and $Y_0$ a RV over $\mathcal{Y}$, define*

$$\bar{\ell}(y_1, y_0, \boldsymbol{x}) := \int_{g^*(y_0|\boldsymbol{x})}^{y_1} h(t, y_0, \boldsymbol{x})\mathrm{d}t \qquad \ell(\boldsymbol{\theta}, y_0, \boldsymbol{x}) := \bar{\ell}\{g_{\boldsymbol{\theta}}(y_0|\boldsymbol{x}), y_0, \boldsymbol{x}\}.$$

$$L(\boldsymbol{\theta}) := \mathbb{E}[\ell(\boldsymbol{\theta}, Y_0, X)] \qquad \tilde{\boldsymbol{\theta}} := \underset{\boldsymbol{\theta} \in \Theta}{\operatorname{argmin}}\, L(\boldsymbol{\theta})$$

In summary, evaluating $\bar{\ell}$ at the CQC estimate, $g_{\boldsymbol{\theta}}(y_0|x)$, yields the pointwise loss, $\ell(\theta, y_0, x)$, whose expectation guides the estimation of $\theta$. Specifically we then have that

$$g^*(y_0|\boldsymbol{x}) = \underset{y_1}{\operatorname{argmin}}\, \bar{\ell}(y_1, y_0, \boldsymbol{x}). \tag{4}$$

This result follows from a simple application of the Fundamental Theorem of Calculus. A detailed proof is provided in Appendix A.1. Now, suppose there exists unique $\boldsymbol{\theta}^* \in \Theta$ such that $g^* = g_{\boldsymbol{\theta}^*}$ and $\operatorname{supp}(Y_0|X = \boldsymbol{x}) = \mathcal{Y}$ for all $\boldsymbol{x} \in \mathcal{X}$ then, as $\boldsymbol{\theta}^*$ minimises $\bar{\ell}\{g_{\boldsymbol{\theta}}(y_0|\boldsymbol{x}), y_0, \boldsymbol{x}\}$ pointwise for all $y_0, \boldsymbol{x}$, we have that $\tilde{\boldsymbol{\theta}} = \boldsymbol{\theta}^*$ i.e. our minimiser is the true parameter.

To further aid in the interpretation and justification of the loss function in Definition 2, including in cases where $\mathcal{G}_{\boldsymbol{\theta}}$ does not contains the true CQC, we will provide various bounds on the loss function in Proposition 1. We do this via three different avenues, each requiring *separate* assumptions on the distribution of our treated response with varying levels of generality. While these bounds are helpful and illustrative, our loss is still justified even when none of these bounds hold.

**Proposition 1.** *For any $y \in \mathcal{Y}$, $\boldsymbol{x} \in \mathcal{X}$, and $\boldsymbol{\theta} \in \Theta$ we have the following upper bound on the loss:*

$$\ell(\boldsymbol{\theta}, y_0, \boldsymbol{x}) \leq |g_{\boldsymbol{\theta}}(y_0|\boldsymbol{x}) - g^*(y_0|\boldsymbol{x})| \, |F_1\{g_{\boldsymbol{\theta}}(y_0|\boldsymbol{x})|\boldsymbol{x}\} - F_1\{g^*(y_1|\boldsymbol{x})|\boldsymbol{x}\}|.$$

*Under various conditions we have the following three lower bounds on the loss:*

(a) *Suppose that $p_1(y|\boldsymbol{x}) \leq \xi_1$ for all $y, \boldsymbol{x}$, then*
$(F_1\{g_{\boldsymbol{\theta}}(y_0|\boldsymbol{x})|\boldsymbol{x}\} - F_1\{g^*(y_0|\boldsymbol{x})|\boldsymbol{x}\})^2 \leq 2\xi_1 \ell(\boldsymbol{\theta}, y_0, \boldsymbol{x}).$

(b) *Suppose that $p_1(y|\boldsymbol{x}) \geq \xi_2$ for all $y, \boldsymbol{x}$, then $\xi_2\{g_{\boldsymbol{\theta}}(y_0|X) - g^*(y_0|X)\}^2 \leq 2\ell(\boldsymbol{\theta}, y_0, \boldsymbol{x}).$*

(c) *Suppose that $p_1(y|\boldsymbol{x})$ is an decreasing function of $y$, then*
$|g_{\boldsymbol{\theta}}(y_0|\boldsymbol{x}) - g^*(y_0|\boldsymbol{x})| \, |F_1\{g_{\boldsymbol{\theta}}(y_0|\boldsymbol{x})|\boldsymbol{x}\} - F_1\{g^*(y_0|\boldsymbol{x})|\boldsymbol{x}\}| \leq 2\ell(\boldsymbol{\theta}, y_0, \boldsymbol{x}).$

The proof is given in Appendix A.1.1.

Error terms involving both $|g_{\boldsymbol{\theta}}(y_0|\boldsymbol{x}) - g^*(y_0|\boldsymbol{x})|$ and $|F_1\{g_{\boldsymbol{\theta}}(y_0|\boldsymbol{x})|\boldsymbol{x}\} - F_0\{g^*(y_0|\boldsymbol{x})|\boldsymbol{x}\}|$ are natural as the first represents the error on our estimator while the second is the error of our estimator when mapped on to probability space. The assumption in (a) covers many common distributions with densities bounded above. The assumption in (b) applies to many bounded-support distributions such as the Beta. The final case is less common but holds for some distributions, e.g., the exponential, and has been studied in density estimation (Birge, 1989).

### 3.1 OUR ESTIMATOR

While the above results justify our loss $\ell$ in Definition 2, they do not give us any approach to evaluate or even approximate it. To achieve this we return back to the derivative of $\bar{\ell}$ (also given in Definition 2) with which we initially motivated our approach. To this end, with $\boldsymbol{z} := (y, \boldsymbol{x}, a)$, define

$$\zeta_{\mathrm{dr}}(\boldsymbol{\theta}, y_0, \boldsymbol{z}) := \nabla_{\boldsymbol{\theta}} g_{\boldsymbol{\theta}}(y_0|\boldsymbol{x}) \left( \frac{a}{\pi(\boldsymbol{x})} \{\mathbb{1}\{y \leq g_{\boldsymbol{\theta}}(y_0|\boldsymbol{x})\} - F_1(g_{\boldsymbol{\theta}}(y_0|\boldsymbol{x})|\boldsymbol{x})\} - \right. \tag{5}$$

$$\left. \frac{1-a}{1-\pi(\boldsymbol{x})} \{\mathbb{1}\{y \leq y_0\} - F_0(y_0|\boldsymbol{x})\} \; + \; F_1(y_1|\boldsymbol{x}) - F_0(y_0|\boldsymbol{x}) \right)$$

$$J(\boldsymbol{\theta}) := \mathbb{E}[\zeta_{\mathrm{dr}}(\boldsymbol{\theta}, Y_0, Z)]. \tag{6}$$

We then have the following proposition.

**Proposition 2.** *For $y_0 \in \mathcal{Y}$, $\boldsymbol{x} \in \mathcal{X}$, and $\boldsymbol{\theta} \in \Theta$ we have that*

$$\mathbb{E}[\zeta_{dr}(\boldsymbol{\theta}, y_0, Z)|X = \boldsymbol{x}] = \nabla_{\boldsymbol{\theta}}\ell(\boldsymbol{\theta}, y_0, \boldsymbol{x}) \; \text{and} \; J(\boldsymbol{\theta}) = \nabla_{\boldsymbol{\theta}}L(\boldsymbol{\theta}).$$

The proof can be found in Appendix A.1.

**Remark 3.** *While an inverse probability weighting approach could instead be used to approximate $\nabla_{\boldsymbol{\theta}}\ell$, this form of $\zeta$ provides the desirable double robustness property, as we will demonstrate later.*

**Remark 4.** *While this only gives us a gradient of a loss function rather than the loss function itself we discuss how an estimate of the loss itself can be derived via 1D quadrature for validation and hyper parameter selection purposes in Appendix B.2.*

This result allows us to use $\zeta_{\mathrm{dr}}$ and samples from $Z$ to perform gradient descent on the sample version of $L(\boldsymbol{\theta})$. In practice, we do not have access to $F_a, \pi$ and so will replace these with estimates given by $\widehat{F}_a, \widehat{\pi}$. We use $\hat{\zeta}_{\mathrm{dr}}$ to represent the version of $\zeta_{\mathrm{dr}}$ with $F_a, \pi$ replaced by $\widehat{F}_a, \widehat{\pi}$. With data, $D = \{Z^{(i)}\}_{i=1}^n$, and testing points $\{Y_0^{(i)}\}$, we define our Monte-Carlo estimate of the gradient to be

$$\hat{J}_{\mathrm{dr}}(\boldsymbol{\theta}, \{(Y_0^{(i)}, Z^{(i)})\}_{i=1}^n) := \frac{1}{n}\sum_{i=1}^n \hat{\zeta}_{\mathrm{dr}}(\boldsymbol{\theta}, Y_0^{(i)}, Z^{(i)}). \tag{7}$$

This finally allows us to define our estimation procedure which is presented in Algorithm 1.

---

**Algorithm 1** Doubly robust, direct CQC estimation algorithm

---

**Require:** $D = \{Z^{(i)}\}_{i=1}^{2n}, \mathcal{G}_\Theta, \boldsymbol{\theta}^{(0)}, T \in \mathbb{N}, \mu > 0$
1: Define $\mathcal{I} := \{1, \ldots, n\}, \mathcal{J} := \{n+1, \ldots, 2n\}$ and split $D$ into $D_{\mathcal{I}} := \{Z^{(i)}\}_{i\in\mathcal{I}}, D_{\mathcal{J}} := \{Z^{(j)}\}_{j\in\mathcal{J}}$.
2: Use $D_{\mathcal{I}}$ to estimate $\hat{\pi}, \widehat{F}_0, \widehat{F}_1$
3: Set $\boldsymbol{\theta} = \boldsymbol{\theta}_0$.
4: **for** $t = 1$ **to** $T$ **do**
5:     For $i \in \mathcal{J}$ sample $Y_0^{(i)}$ (potentially dependent upon $X^{(i)}$). See Remark 6 for more detail.
6:     Obtain our Monte-Carlo estimate $J(\boldsymbol{\theta}^{(t)})$ given by $\hat{J}_{\mathrm{dr}}(\boldsymbol{\theta}, \{(Y_0^{(i)}, Z^{(i)})\}_{i\in\mathcal{J}})$ in (7).
7:     Update $\boldsymbol{\theta}$ by $\boldsymbol{\theta}^{(t+1)} = \boldsymbol{\theta}^{(t)} - \mu\hat{J}(\boldsymbol{\theta}^{(t)})$.
8: **end for**
9: **return** $\boldsymbol{\theta}^{(T)}$.

---

**Remark 5.** *In practice, we can replace step 7 of Algorithm 1 with any exclusively gradient-based (stochastic or otherwise) optimisation procedure such as Adam (Kingma and Ba, 2015).*

**Remark 6.** *We can choose our distribution over $Y_0$ relatively flexibly as this simply defines the test points for our CQC function (similarly to choosing the quantile level $\alpha$ in CQTE estimation). We commonly take $Y_0 \sim Y|A = 0$ with $Y_0 \perp Z$ by simply choosing random untreated responses for each sample. Thus testing our CQC at typical $Y_0$ values. An experiment testing this choice is given in Appendix D.5.*

Due to its more direct nature, this estimation procedure solves all three problems of the previous inversion approach discussed in Section 2.2. Crucially, its explicit parameterisation of the CQC allows us to inform and constrain our model, as well as making our model more interpretable and significantly faster to sample from. In addition, since the estimation procedure operates directly on $g_{\boldsymbol{\theta}}$, we might naturally suspect its accuracy to depend upon the complexity of the underlying CQC. We might also hope it retains the double-robustness property present in the previous approach. Below, we show that both of these properties hold.

## 3.2 ACCURACY RESULTS

As we intend to use gradient descent for our minimisation, a natural question is when is this procedure guaranteed to converge and at what rate does this convergence occur. We now make some restrictions on our model architecture which allow us to achieve this.

**Assumption 1.** *For all $y_0 \in \mathcal{Y}, \boldsymbol{x} \in \mathcal{X}, \boldsymbol{\theta} \in \Theta$:*

*(a) $a < \widehat{\pi}(\boldsymbol{x}) < 1 - a$ for some $a > 0$.*

*(b) $g_{\boldsymbol{\theta}}$ is of the form $g_{\boldsymbol{\theta}}(y_0|\boldsymbol{x}) = \boldsymbol{\theta}^\top \boldsymbol{f}(y_0, \boldsymbol{x})$ for some feature function $\boldsymbol{f} : \mathcal{Y} \times \mathcal{X} \to \mathbb{R}^p$.*

(c) $\|\boldsymbol{f}(y_0, \boldsymbol{x})\| \le \rho$ for some $\rho > 0$.

Assumption 1(a) assumes that we can bound our estimated propensity away from $\{0, 1\}$, this is a common assumption within HTE literature and is not very restrictive due to the true propensity already being assumed to be bounded away from $\{0, 1\}$. Assumption 1(b) enforces convexity of our loss function w.r.t. $\boldsymbol{\theta}$ and bears similarity to the linear smoother framework used in Kallus and Oprescu (2023); Kennedy (2023a). Importantly, this assumption does not confine us to linear CQC functional estimates as the form of $\boldsymbol{f}$ can be chosen freely, enabling the use of kernel methods via random Fourier features (Avron et al., 2017; Liu et al., 2022; Rahimi and Recht, 2007) and other general architectures. Assumption 1(c) is required in order to control the rate at which our CQC estimate changes with respect to our parameter $\boldsymbol{\theta}$.

**Theorem 3.** *Let $\tilde{\boldsymbol{\theta}}$ be the minimiser of our population loss as given in Definition 2. Suppose that Assumption 1 holds and that $\|\tilde{\boldsymbol{\theta}}\| \le B$ for some $B > 0$. For $t \in [n]$, define $\boldsymbol{\theta}^{(t)}$ inductively by $\boldsymbol{\theta}^{(1)} = \boldsymbol{0}$, $\boldsymbol{\theta}^{(t+\frac{1}{2})} = \boldsymbol{\theta}^{(t)} - \mu_t v^{(t)}$, and $\boldsymbol{\theta}^{(t+1)} = \arg\min_{\boldsymbol{\theta} : \|\boldsymbol{\theta}\| \le B} \|\boldsymbol{\theta} - \boldsymbol{\theta}^{(t+\frac{1}{2})}\|$, with, $\mu_t = \frac{Bc}{2\rho\sqrt{n}}$, and $v^{(t)} := \hat{\zeta}(\boldsymbol{\theta}^{(t)}, Y_0^{(t)}, Z^{(t)})$. Finally, define our parameter estimate as $\hat{\boldsymbol{\theta}} = \frac{1}{n} \sum_{t=1}^{n} \boldsymbol{\theta}^{(t)}$. Then, if $\hat{\pi}, \widehat{F}_a$ are independent of $\left\{ \left( Y_0^{(t)}, Z^{(t)} \right) \right\}_{t=1}^{n}$, we have that*

$$\mathbb{E}[L(\hat{\boldsymbol{\theta}}) - L(\tilde{\boldsymbol{\theta}})] \le C_1 \left( 1/\sqrt{n} + \varepsilon(\hat{\pi}, \widehat{F}_0, \widehat{F}_1) \right) \quad \text{with} \tag{8}$$

$$\varepsilon(\hat{\pi}, \widehat{F}_0, \widehat{F}_1) := \sqrt{\mathbb{E}\left[ \left( \pi(X) - \hat{\pi}(X) \right)^2 \right] \mathbb{E}\left[ \sup_{y_0 \in \mathcal{Y}, a \in \{0,1\}} \left( F_a(y_0|X) - \widehat{F}_a(y_0|X) \right)^2 \right]} \tag{9}$$

*where $C_1$ is a constant depending upon, $B, c, \rho$. Suppose further that the assumption in Proposition 1(b) holds and that $\mathbb{E}[\boldsymbol{f}(Y_0, X)\boldsymbol{f}(Y_0, X)^\top] \ge \eta_2$. If we instead take $\mu_t = \frac{1}{\xi_2 \eta_2 n}$ then we have that*

$$\mathbb{E}[L(\hat{\boldsymbol{\theta}}) - L(\tilde{\boldsymbol{\theta}})] \le C_2 \left( \log(n)/n + \varepsilon(\hat{\pi}, \widehat{F}_0, \widehat{F}_1) \right) \tag{10}$$

*where $C_2$ is a constant depending upon, $B, c, \rho, \xi_2, \eta_2$.*

The proof is provided in Appendix A.2. An additional result giving high probability bounds of the same rate as (8) is given by Proposition 11 in Appendix A.2.4. The requirement for the nuisance parameter estimates to be independent of the data used for fitting the CQC motivates the sample-splitting procedure in Algorithm 1. One could instead use a cross-fitting approach after sample-splitting and average the two CQC estimates which would lead to comparable theoretical results.

Regarding the result, first we see that in both (8) & (10) we have *double robustness*. This is because both of the nuisance parameter estimators can converge *slower* than the leading term in the error while not affecting the overall convergence rate due to said errors multiplying. This is similar to other doubly robust approaches which have been presented for the CATE (Kennedy, 2023b), CQTE (Kallus and Oprescu, 2023), and CQC (Givens et al., 2024) which all also derive their robustness results via a product of errors over the nuisance parameters. For the second result our requirement on the nuisance parameter estimation is stronger however as we need to obtain $\log(n)/n$ convergence on the product of the nuisance parameter estimates.

We also note that when our density is bounded below as in the second result, if $\tilde{\boldsymbol{\theta}} = \boldsymbol{\theta}^*$ where $g_{\boldsymbol{\theta}^*} = g^*$, we have (using Proposition 1) that $\mathbb{E}[\{g_{\hat{\boldsymbol{\theta}}}(Y_0|X) - g^*(Y_0|X)\}^2] \le \mathbb{E}[L(\hat{\boldsymbol{\theta}}) - L(\tilde{\boldsymbol{\theta}})]$. Hence if the nuisance term converges at the same rate as the leading term we get a convergence rate on the root mean square error (RMSE) of our CQC estimate of order $1/\sqrt{n}$ which is desirable. Furthermore from Assumption 1 (c) this gives convergence of $\hat{\boldsymbol{\theta}}$ to $\tilde{\boldsymbol{\theta}}$ of $1/\sqrt{n}$ as well.

## 4 SIMULATED RESULTS

We now illustrate the advantages of our approach by comparing it to two alternatives across multiple dimensions. First, we evaluate it against the previously proposed inverting CQC estimation method from Givens et al. (2024) (labelled "Inv. DR") to highlight the benefits of our direct CQC parameterisation. Second, we compare it to an inverse probability weighting (IPW) variant of our method, where $\zeta_{\text{dr}}$ is replaced by its IPW counterpart (labelled "IPW"; see Appendix B.1 for details), to demonstrate the gains from our double robustness. For each method we present an oracle version

which uses the exact nuisance parameters $(F_a, \pi)$ and an estimated version that uses their estimated equivalents. Further details can be found in Appendix C.2. Further comparisons to the S-Learner approach, where $\widehat{F}_0, \widehat{F}_1$ are used to directly produce our CQC estimate are given in Appendix D.

Throughout each experiment, we take $X \sim N(0, I_d)$ for $d = 10$, $Y|X = \boldsymbol{x}, A = a \sim N(\sin(\pi \boldsymbol{v}^\top \boldsymbol{x}) + a\gamma \boldsymbol{v}^\top \boldsymbol{x}, 1)$ and $\pi(\boldsymbol{x}) = \sigma(\boldsymbol{v}^\top \boldsymbol{x})$ where $\boldsymbol{v}$ is a random vector in $\mathbb{R}^d$ with $\|\boldsymbol{v}\| = \sqrt{d}$, $\sigma$ is the sigmoid function, and $\gamma > 0$ can be varied. The sine term represents complexity in the marginal distributions as this an oscillating nonlinear change in the distribution. We thus have that the CCDFs contain the oscillating sine dependency over $\boldsymbol{x}$ while the CQC itself does not, simply being $g^*(y_0|\boldsymbol{x}) = \gamma \boldsymbol{v}^\top \boldsymbol{x}$.

We test two distinct models for the CQC. The first, "DR-Lin", is a correctly specified linear model where we take $g_{\boldsymbol{\theta}}(y_0|\boldsymbol{x}) = (\boldsymbol{\theta}_{\mathrm{sc}}^\top \boldsymbol{x} + \theta_{\mathrm{sc},0})(y_0) + (\boldsymbol{\theta}_{\mathrm{sh}}^\top \boldsymbol{x} + \theta_{\mathrm{sh},0})$ so that $\boldsymbol{\theta}_{\mathrm{sc}}, \boldsymbol{\theta}_{\mathrm{sh}}$ represent the scaled and shift components of the CQC respectively. The second, "DR-NN" is a full connected Neural Network (NN) with ReLU activations and 2 hidden layers each of width 20.

We fit the propensity score via logistic regression and the CCDFs using kernel CCDF estimation in order to effectively model the sine terms. For each of the following experiments, 100 runs are repeated and mean absolute error of our CQC estimate alongside 95% confidence intervals are presented. Code to reproduce all experiments is provided in the Supplementary Materials. Further experiments with different distributional settings are given in Appendix D.1 and experiments exploring sensitivity of performance to hyperparameters are given in Appendix D.3.[1]

## 4.1 INCREASING STEEPNESS OF THE CQC

For the first experiment we increase $\gamma$ to increase the slope of the CQC. As our current approach is able to model the CQC directly as a linear function, it should be minimally affected by the increase in slope while methods which cannot model this linearity will struggle. Figure 2a shows that our directly parameterised approach (Est. DR-Lin) does indeed perform stronger especially at larger slopes. We see that our NN approach also performs comparably to the linear model. While our estimated versions (Est. DR-Lin/NN) are somewhat worse than their oracle counterparts, they still outperform the oracle inverting method.

## 4.2 INCREASING THE ERROR OF NUISANCE PARAMETERS

We further investigate how errors in nuisance parameter estimation affect our estimator's accuracy. To do this, we add increasing levels of biased, random noise to the logits of the original nuisance parameter estimates. Results are shown in Figure 2b. We observe that both parameterisations of our method (Est. DR-Lin, Est. DR-NN) perform strongest with the linear model performing marginally better. We also see that the inverting approach (Est. Inv. DR) performs well under increasing nuisance parameter error. Interestingly, the inverting estimator appears somewhat less sensitive to this error than our approach. Nonetheless, our gradient-based approaches (Est. DR-Lin/NN) perform comparably or better across almost all levels of added noise. Additional experiments estimating each nuisance parameter separately is given in Appendix D.4.

## 4.3 INCREASING SAMPLE SIZE

Finally we plot the error of these estimation procedures over various sample sizes which can be found in Figure 2c. We can see that, once again, our approach performs best, achieving the lowest mean error across all sample sizes and demonstrating consistent improvement as sample size increases.

To summarise, across all our results we see that our approach is the strongest for both a linear and NN based CQC model with substantial gains over the existing inverting approach especially when the slope of the CQC is larger. We see that the linear CQC model is marginally stronger than the NN model throughout which we would expect due to it encompassing the true CQC while being a simpler model. Overall, these results are promising as they suggest that not only is our approach strong, but it maintains much of this strength even when we do not know the explicit parametric form of the CQC.

## 5 REAL WORLD SETTING

We also apply our results to real world data to demonstrate their interpretability. Here we look at an employment example which has been studied in multiple heterogeneous treatment effect examples

---

[1]All code and data for the experiments presented can be found at `https://github.com/joshgivens/DirectEstimationofConditionalQuantileContrasts`.

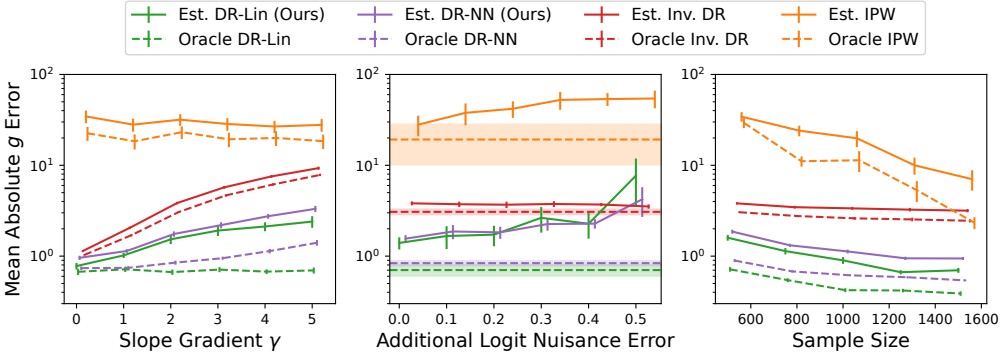

(a) Varying CQC slope steepness w.r.t. $\boldsymbol{x}$ with sample size 500.

(b) Varying nuisance parameter error with sample size 500 and $\gamma = 2$.

(c) Varying sample size with $\gamma = 2$

Figure 2: Mean absolute error of CQC estimate for various methods with 95% C.I.s over 100 runs.

(Autor and Houseman, 2010; Autor et al., 2017; Powell, 2020; Givens et al., 2024). Here, the intervention ($A = 1$) corresponds to enrolment in an employment programme, and the outcome ($Y$) represents total earnings in a two-year period in thousands of dollars.

For our estimation, we use the linear CQC model described in Section 4. We then subtract $y_0$ from $\hat{g} = g_{\hat{\boldsymbol{\theta}}}$, to estimate $\Delta(y_0|\boldsymbol{x}) := g^*(y_0|\boldsymbol{x}) - y_0$. This enables easier interpretation as positive and negative values of $\Delta$ are associated with benefit and detriment of the intervention respectively.

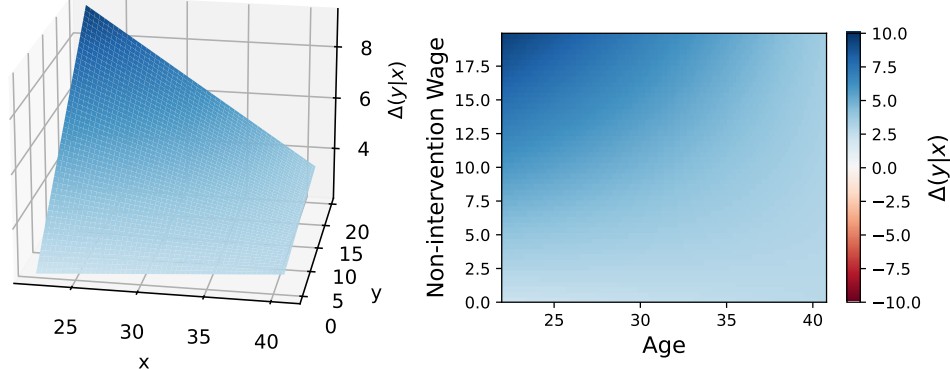

Figure 3: Surface and heat plot of $\Delta^*(y|\boldsymbol{x})$ for our employment data with $X =$Age, $Y=$Income.

Figure 3 shows this estimate for various values of $(y, \boldsymbol{x})$. From these results we see an interesting pattern. Across all ages, the intervention had the most impact for those with high non-intervention earnings. The change in wage improvement as a function of non-intervention wages seems to decrease as age increases however. In other words for younger participants, the distribution of wages seems to multiplicatively scale while for older participants, the impact of treatment seems to be better represented by a more uniform shift. We examine the parameters of our estimate directly in Appendix D.6. Another example examining the effect of a treatment on colon cancer remission is presented in Appendix D.7 where we use a neural network (NN) to model a nonlinear CQC function.

## 6 LIMITATIONS AND FUTURE WORK

One limitation of our approach is that while our direct estimator performs best overall, there is evidence to suggest it is practically more sensitive to nuisance parameter estimation error than the existing inversion based estimation approach. This is somewhat mirrored in Theorem 3, where our double robustness is with respect to error on our loss function rather than directly on error of the CQC. Future work could investigate these two properties and their relationship more thoroughly, with the potential to improve upon them further.

Additionally, while our estimator is direct in terms of exclusively estimating our estimand of interest, it does not have the form of estimating the estimand through a conditional expectation as is common for other estimators (e.g. Kennedy (2023b); Kallus and Oprescu (2023).) Such an estimator then has the advantage of being estimable by various non-parametric procedures for conditional expectation estimation while also being estimable parametrically via least squares. It also has the advantage of giving accuracy results directly in terms of the estimand of interest which we are only able to do under certain settings. As such, a future direction would be to explore whether a doubly robust estimator of this form could be produced for the CQC.

Finally, while our current convergence results apply to a good number of parametric and nonparametric CQC models, later work could expand these results to CQC estimates which are not linear with respect to their parameters, such as NNs (Shi et al., 2019) or Bayesian additive regression trees (Hill, 2011; Green and Kern, 2012; Künzel et al., 2019).

## 7 CONCLUSION

To conclude, we have proposed the first direct estimation procedure for the CQC, an estimand which aims to bridge the gap between the CATE and the CQTE. We have demonstrated the efficacy of this new estimation procedure both theoretically and empirically, showing it to outperform existing approaches. Furthermore, we have highlighted its ability to allow for direct parameterisation of the CQC and demonstrated its benefit in terms of both empirical performance and interpretability in real-world scenarios. Overall, this represents an improvement over existing CQC methods, further enhancing the utility and real-world applicability of this emerging treatment effect estimand.

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

# A ADDITIONAL THEORY AND PROOFS

## A.1 LOSS JUSTIFICATION PROOFS

*Proof of equation* (4). As a reminder the identity of interest is

$$g^*(y_0|\boldsymbol{x}) = \underset{y_1}{\operatorname{argmin}}\, \bar{\ell}(y_1, y_0, \boldsymbol{x}).$$

First define the intermediary loss function

By the fundamental theorem of calculus we have that $\partial_{y_1}\bar{\ell}(y_1, y_0, \boldsymbol{x}) = h(y_1, y_0, \boldsymbol{x})$. Therefore, as $\partial_{y_1}\bar{\ell}(y_1, y_0, \boldsymbol{x})$ is increasing in $y_1$ for any $\boldsymbol{x}, y_0$, we have that

$$y_1 = \underset{y_1'}{\operatorname{argmin}}\, \bar{\ell}(y_1', y_0, \boldsymbol{x}) \iff \partial_{y_1}\bar{\ell}(y_1, y_0, \boldsymbol{x}) = 0$$

$$\iff F_1(y_1|\boldsymbol{x}) - F_0(y_0|\boldsymbol{x}) = 0$$
$$\iff F_1(y_1|\boldsymbol{x}) = F_0(y_0|\boldsymbol{x}).$$

Hence by definition of $g^*$, we have that

$$g^*(y_0|\boldsymbol{x}) = \underset{y_1}{\operatorname{argmin}}\, \bar{\ell}(y_1, y_0|\boldsymbol{x}).$$

$\square$

*Proof of Proposition 2.* Firstly define $\bar{\zeta}(y_1, y_0|\boldsymbol{x})$ by

$$\bar{\zeta} := \frac{a}{\pi(\boldsymbol{x})}\left\{\mathbb{1}\{y \le g_{\boldsymbol{\theta}}(y_0|\boldsymbol{x})\} - F_1(g_{\boldsymbol{\theta}}(g|\boldsymbol{x})|\boldsymbol{x})\right\} - \frac{1-a}{1-\pi(\boldsymbol{x})}\left\{\mathbb{1}\{y \le y_0\} - F_0(y_0|\boldsymbol{x})\right\}$$
$$+ F_1(y_1|\boldsymbol{x}) - F_0(y_0|\boldsymbol{x}).$$

So that $\zeta(\boldsymbol{\theta}, y_0, \boldsymbol{x}) = \{\nabla_{\boldsymbol{\theta}} g_{\boldsymbol{\theta}}(y_0|\boldsymbol{x})\}\bar{\zeta}(g_{\boldsymbol{\theta}}(y_0|\boldsymbol{x}), y_0, \boldsymbol{x})$. By the chain rule we have that $\nabla_{\boldsymbol{\theta}}(\boldsymbol{\theta}, y_0, \boldsymbol{x}) = \nabla_{\boldsymbol{\theta}} g_{\boldsymbol{\theta}}(y_0|\boldsymbol{x})\partial_{g_{\boldsymbol{\theta}}(y_0|\boldsymbol{x})} \cdot \ell(g_{\boldsymbol{\theta}}(y_0|\boldsymbol{x}), y_0|\boldsymbol{x})$.

Hence all that is left to show is that $\mathbb{E}[\bar{\zeta}(y_1, y_0, Z)|X = \boldsymbol{x})]\partial_{y_1}\bar{\ell}(y_1, y_0|\boldsymbol{x})$. To this end we can use the tower property to get that

$$
\begin{aligned}
\mathbb{E}[\bar{\zeta}_{\mathrm{dr}}(y_1, y_0, Z)|X = \boldsymbol{x}] = \; & \mathbb{E}\left[\frac{A}{\pi(\boldsymbol{x})}\underbrace{\{\mathbb{E}[\mathbb{1}\{Y \le y_1\}|X, A = 1] - F_1(y_1|\boldsymbol{x})\}}_{=0}\right] \\
& - \mathbb{E}\left[\frac{1-A}{1-\pi(\boldsymbol{x})}\underbrace{\{\mathbb{E}[\mathbb{1}\{Y \le y_0\}|X, A = 0] - F_0(y_0|\boldsymbol{x})\}}_{=0}\right] \\
& + F_1(y_1|\boldsymbol{x}) - F_0(y_0|\boldsymbol{x}) \\
= \; & F_1(y_1|\boldsymbol{x}) - F_0(y_0|\boldsymbol{x}) \\
= \; & \partial_{y_1}\bar{\ell}(y_1, y_0, \boldsymbol{x}).
\end{aligned}
$$

$\square$

### A.1.1 Loss bound proofs

We now provide the proof for our result bounding the loss in various circumstances. First however we provide a proposition with various upper and lower bounds on integrals which will inform our upper and lower bounds on the loss.

**Proposition 4.** *Let $F : \mathcal{Y} \to \mathbb{R}$ be an arbitrary increasing function with $F(a) = 0$, $F(b) = \beta$ for $a < b \in \mathcal{Y}$. Also define $f(y) = \partial_y F(y)$ and $I = \int_a^b F(y)\mathrm{d}y$. We then have that*

1. *$I \le |\beta|\,|b-a|$.*
2. *If $f(y) \ge \eta$ for all $y \in [a, b]$, $I \ge \frac{\eta}{2}(b-a)^2$.*
3. *If $f(y) \le \xi$ for all $y \in [a, b]$, $I \ge (2\xi)^{-1}\beta^2$.*
4. *If $f(y)$ is increasing on $[a, b]$ then $I \ge \frac{1}{2}|\beta|\,|b-a|$.*

*As a convention we allow for the possibility that $a > b$ and take $[a, b]$ in this case to mean $[b, a]$.*

*Proof.* All results are proved under the case $a \le b$. The results for the case $a > b$ follow an identical argument with signs and equalities reversed. The first result follows directly from the fact that $F(y) \le F(b)$ for all $y \in [a, b]$.

For the second result we have that

$$
\begin{aligned}
F(y) &= \int_a^y f(s)\mathrm{d}s + F(a) \\
&= \int_a^y f(s)\mathrm{d}s \\
&\ge (y-a)\eta.
\end{aligned}
$$

Therefore

$$
\begin{aligned}
I &\ge \int_a^b (y-a)\eta\mathrm{d}y \\
&= \frac{\eta}{2}(b-a)^2.
\end{aligned}
$$

For the third result define $\tilde{F} : [a, b] \to \mathcal{Y}$ by

$$
\tilde{F}(y) = \begin{cases} 0 & \text{if } y \in [a, b - \beta/\xi], \\ \xi y - \xi b + \beta & \text{if } y \in (b - \beta/\xi, b]. \end{cases}
$$

Then $\tilde{F}$ is, non-negative, continuous and increasing with $F(b) = \beta$ and maximum gradient $\xi$. Furthermore we claim that $\tilde{F}$ lower bounds any other functions with this property which also has continuous derivative.

This is trivially true for $y \in [a, b - \beta/\xi]$. Otherwise suppose there exists function $G$ satisfying all these assumptions excluding the gradient bound with $G(y) < \tilde{F}(y)$ for some $y \in (b - \beta/\xi, b]$. Then we have that $\frac{G(b) - G(y)}{b - y} < \xi$, hence by the mean value theorem we must have that $\partial_y G(y') < \xi$ for some $y'$ in $[y, b]$. Thus by the contrapositive, $\tilde{F}$ is the minimal function satisfying all these conditions on $(b - \beta/\xi, b]$.

As such we can now get the following bound on $I$

$$I \geq \int_a^b \tilde{F}(y)\mathrm{d}y$$

$$= \int_{b-\beta/\xi}^b \xi y - \xi b + \beta \mathrm{d}y = \beta^2/\xi.$$

For the final result note that $f(y)$ increasing implies that $F(y)$ is convex. Therefore we have that

$$I = \int_a^b F(y)\mathrm{d}y$$

$$\geq \int_a^b F(a) + \left( \frac{y - a}{b - a} \left( F(b) - F(a) \right) \right) \mathrm{d}y$$

$$\leq \int_a^b \frac{y - a}{b - a} \beta \mathrm{d}y = \frac{1}{2} |\beta| \, |b - a| \, .$$

$\square$

*Proof of Proposition 1.* For notational convenience we introduce the function $h : \mathcal{Y} \times \mathcal{Y} \times \mathcal{X} \to [-1, 1]$ given by

$$h(y_1, y_0, \boldsymbol{x}) := F_1(y_1|\boldsymbol{x}) - F_0(y_0|\boldsymbol{x})$$

so that $\bar{\ell}(y_1, y_0, \boldsymbol{x}) := \int_{g^*(y_0|\boldsymbol{x})}^{y_1} h(t, y_0, \boldsymbol{x})\mathrm{d}t$. Remember that $\ell(\boldsymbol{\theta}, y_0, \boldsymbol{x}) = \bar{\ell}(g_{\boldsymbol{\theta}}(y_0|\boldsymbol{x}), y_0, \boldsymbol{x})$. We can then notice that $h(y_1, y_0, \boldsymbol{x})$ satisfies the conditions of Proposition 4 as a function of $y_1$ with $a = g^*(y_0, \boldsymbol{x})$ and $b = g_{\boldsymbol{\theta}}(y_0|\boldsymbol{x})$ and $\beta = F_1(g_{\boldsymbol{\theta}}(y_0|\boldsymbol{x})|\boldsymbol{x}) - F_0(y_0|\boldsymbol{x})$.

Furthermore $\partial_{y_1} \bar{\ell}(y_1, y_0, \boldsymbol{x}) = p_1(y_1|\boldsymbol{x})$. Therefore the assumptions in Proposition 1(a)-(c) correspond to the assumptions in results 2-4 of Proposition 4.

Therefore we can simply directly apply each result of Proposition 4 prove our required results. $\square$

## A.2 ESTIMATION ACCURACY THEORY AND PROOFS

### A.2.1 CONVEX CONVERGENCE

**Lemma 5.** *Let $L(\boldsymbol{\theta})$ be a convex function and define $\tilde{\boldsymbol{\theta}} = \mathrm{argmin}_{\boldsymbol{\theta}} L(\boldsymbol{\theta})$ with $\|\tilde{\boldsymbol{\theta}}\| \leq B$ for some $B > 0$. Define $\boldsymbol{\theta}^{(1)} = \boldsymbol{0}$ and inductively take*

$$\boldsymbol{\theta}^{(t+\frac{1}{2})} = \boldsymbol{\theta}^{(t)} - \eta v^{(t)} \qquad\qquad \boldsymbol{\theta}^{(t+1)} = \mathrm{argmin}_{\boldsymbol{\theta}:\|\boldsymbol{\theta}\|\leq B} \|\boldsymbol{\theta} - \boldsymbol{\theta}^{(t+\frac{1}{2})}\|$$

*with $\eta = \frac{B}{\tilde{\rho}\sqrt{n}}$ and $v_1, \ldots, v_n$ a sequence of RVs with $\|v^{(t)}\| \leq \tilde{\rho}$. Finally, take our parameter estimate to be $\hat{\boldsymbol{\theta}} = \frac{1}{n} \sum_{t=1}^n \boldsymbol{\theta}^{(t)}$.*

*Then we have that*

$$L(\hat{\boldsymbol{\theta}}) - L(\tilde{\boldsymbol{\theta}}) \leq \frac{B\tilde{\rho}}{\sqrt{n}} - \frac{1}{n} \sum_{t=1}^n \left\langle \boldsymbol{\theta}^{(t)} - \tilde{\boldsymbol{\theta}}, \varepsilon^{(t)} \right\rangle$$

*where $\varepsilon^{(t)} := v^{(t)} - \nabla_{\boldsymbol{\theta}^{(t)}} L(\boldsymbol{\theta}^{(t)})$.*

*Proof of Lemma 5.* Define $\nabla^{(t)} := \nabla_{\boldsymbol{\theta}^{(t)}} L(\boldsymbol{\theta}^{(t)})$ so that $\mathbb{E}[v^{(t)}|\boldsymbol{\theta}^{(t)}] = \nabla^{(t)} + \varepsilon^{(t)}$ where $\nabla^{(t)}$ represents the unbiased gradient estimate and $\varepsilon^{(t)}$ represents the bias.

From Shalev-Shwartz and Ben-David (2014) section 14.4.1 we have that

$$\mathbb{E}\left[\frac{1}{n}\sum_{t=1}^{n}\left\langle\boldsymbol{\theta}^{(t)}-\boldsymbol{\theta}^{*},v^{(t)}\right\rangle\right] \leq \frac{B\tilde{\rho}}{\sqrt{n}}$$

Additionally we have

$$
\begin{aligned}
L(\hat{\boldsymbol{\theta}}) - L(\boldsymbol{\theta}^{*}) &\leq \frac{1}{n}\sum_{t=1}^{n}L(\boldsymbol{\theta}^{(t)}) - L(\boldsymbol{\theta}^{*}) \quad \text{by Jensen's inequality.} \\
&\leq \frac{1}{n}\sum_{t=1}^{n}\left\langle\boldsymbol{\theta}^{(t)}-\boldsymbol{\theta}^{*},\nabla^{(t)}\right\rangle \quad \text{by convexity of } L \text{ and definition of } \nabla^{(t)} \\
&= \frac{1}{n}\sum_{t=1}^{n}\left\langle\boldsymbol{\theta}^{(t)}-\boldsymbol{\theta}^{*},v^{(t)}\right\rangle - \frac{1}{n}\sum_{t=1}^{n}\left\langle\boldsymbol{\theta}^{(t)}-\boldsymbol{\theta}^{*},\varepsilon^{(t)}\right\rangle \\
&\leq \frac{B\rho}{\sqrt{n}} - \frac{1}{n}\sum_{t=1}^{n}\left\langle\boldsymbol{\theta}^{(t)}-\boldsymbol{\theta}^{*},\varepsilon^{(t)}\right\rangle \quad \text{from our prior result}
\end{aligned}
$$

$\square$

**Lemma 6.** *Suppose that assumption 1 holds. For arbitrary fixed $\boldsymbol{\theta} \in \Theta$, Define*
$$\varepsilon = \zeta(\boldsymbol{\theta}, Y_0, Z) - \nabla_{\boldsymbol{\theta}}L(\boldsymbol{\theta})$$

*Then we have that*

$$\|\mathbb{E}[\varepsilon]\| \leq \frac{2\rho}{c}\sqrt{\mathbb{E}\left[\left|\left|\pi(X)-\widehat{\pi}(X)\right|\right|^2\right]\mathbb{E}\left[\sup_{\substack{y_0\in\mathcal{Y},\\a\in\{0,1\}}}\left|F_a(y_0|X)-\widehat{F}_a(y_0|X)\right|^2\right]}.$$

*Proof.* To do this first define

$$\hat{b}(\boldsymbol{\theta}, Y_0, X) := \mathbb{E}[\hat{\zeta}(\boldsymbol{\theta}, Y_0, Z) - \zeta(\boldsymbol{\theta}, Y_0, Z)|X, Y_0, \boldsymbol{\theta}]$$

To bound $\hat{b}(\boldsymbol{\theta}, Y_0, Z)$ firstly have that
$$
\begin{aligned}
\mathbb{E}[\mathbb{1}\{Y\leq y\}\mathbb{1}\{A=a\}|X] &= \mathbb{E}[\mathbb{1}\{Y\leq y\}|A=a]\mathbb{P}(A=a|X) \\
&= F_a(y|X)\mathbb{P}(A=a|X).
\end{aligned}
$$
We can then use the fact that $\mathbb{P}(A=1|X) = \pi(X)$ to get

$$
\begin{aligned}
\mathbb{E}[\hat{\zeta}(\boldsymbol{\theta},Y_0,Z)|\boldsymbol{\theta},Y_0,X] = \nabla_{\boldsymbol{\theta}}g_{\boldsymbol{\theta}}(Y_0|X)\bigg\{ &\left(\frac{\pi(X)}{\widehat{\pi}(X)}\right)\left(F_1\{g_{\boldsymbol{\theta}}(Y_0|X)|X\}-\widehat{F}_1\{g(Y_0|X)|X\}\right) \\
&- \frac{1-\pi(X)}{1-\widehat{\pi}(X)}\left(F_0\{y_0|X\}-\widehat{F}_0\{Y_0|X\}\right) \\
&+ \widehat{F}_1\{g_{\boldsymbol{\theta}}(Y_0|X)|X\}-\widehat{F}_0(Y_0|X)\bigg\}.
\end{aligned}
$$

Hence

$$
\begin{aligned}
\hat{b}(\boldsymbol{\theta},Y_0,X) = \nabla_{\boldsymbol{\theta}}g_{\boldsymbol{\theta}}(Y_0|X)\bigg\{ &\left(\frac{\pi(X)}{\widehat{\pi}(X)}-1\right)\left(F_1\{g_{\boldsymbol{\theta}}(Y_0|X)|X\}-\widehat{F}_1\{g_{\boldsymbol{\theta}}(Y_0|X)|X\}\right) \\
&- \left(\frac{1-\pi(X)}{1-\widehat{\pi}(X)}-1\right)\left(F_0(Y_0|X)-\widehat{F}_0(Y_0|X)\right)\bigg\}.
\end{aligned}
$$

Now by the tower property and linearity of expectation, we have that $\mathbb{E}[\varepsilon] = \mathbb{E}[\hat{b}(\boldsymbol{\theta}, Y_0, X)|\boldsymbol{\theta}]$. In turn we then get

$$\|\mathbb{E}[\varepsilon]\| \le \mathbb{E}\left[\left\|\hat{b}(\boldsymbol{\theta}^{(t)}, Y_0, X)\right\| |\boldsymbol{\theta}^{(t)}\right] \quad \text{by Jensen's inequality.}$$

Now using our bound on $\boldsymbol{f}$ in assumption 1, we get that $\|\nabla_{\boldsymbol{\theta}} \boldsymbol{f}(y_0, \boldsymbol{x})\| \le \rho$ for all $y, \boldsymbol{x}$. Additionally using our bound on $\widehat{\pi}$ we get that

$$\left|\frac{1-\pi(\boldsymbol{x})}{1-\widehat{\pi}(\boldsymbol{x})} - 1\right| = \left|\frac{\pi(\boldsymbol{x})}{\widehat{\pi}(\boldsymbol{x})} - 1\right| \le \frac{|\pi(\boldsymbol{x}) - \widehat{\pi}(\boldsymbol{x})|}{c}$$

Combining these we get

$$\mathbb{E}[\|\varepsilon\|] \le \frac{\rho}{c}\mathbb{E}\left[\left|\left(\pi(X) - \widehat{\pi}(X)\right)\right.\right.$$

$$\left.\left.\left(F_1\{g_{\boldsymbol{\theta}}(Y_0|X)|X\} - \widehat{F}_1\{g_{\boldsymbol{\theta}}(Y_0|X)|X\} + F_0(Y_0|X) - \widehat{F}_0(Y_0|X)\right)\right|\right]$$

$$\le \frac{2\rho}{c}\sqrt{\mathbb{E}\left[\left|\pi(X) - \widehat{\pi}(X)\right|^2\right]\mathbb{E}\left[\sup_{\substack{y_0 \in \mathcal{Y}, \\ a \in \{0,1\}}}\left|F_a(y_0|X) - \widehat{F}_a(y_0|X)\right|^2\right]}.$$

$\square$

**Proposition 7.** *Suppose that assumption 1 holds and that $\|\tilde{\boldsymbol{\theta}}\| \le B$ for some $B > 0$. For $t \in [n]$, define $\boldsymbol{\theta}^{(t)}$ inductively by*

$$\boldsymbol{\theta}^{(t+\frac{1}{2})} = \boldsymbol{\theta}^{(t)} - \mu_t v^{(t)} \qquad\qquad \boldsymbol{\theta}^{(t+1)} = \underset{\boldsymbol{\theta}:\|\boldsymbol{\theta}\|\le B}{\operatorname{argmin}}\|\boldsymbol{\theta} - \boldsymbol{\theta}^{(t+\frac{1}{2})}\|$$

*with $\boldsymbol{\theta}^{(1)} = \boldsymbol{0}$, $\mu_t = \frac{Bc}{2\rho\sqrt{n}}$, and $v^{(t)} := \hat{\zeta}(\boldsymbol{\theta}^{(t)}, Y_0^{(t)}, Z^{(t)})$. Finally, define the parameter estimate as $\hat{\boldsymbol{\theta}} = \frac{1}{n}\sum_{t=1}^n \boldsymbol{\theta}^{(t)}$. Then, if $\widehat{\pi}, \widehat{F}_a$ are independent of $\left\{(Y_0^{(t)}, Z^{(t)})\right\}_{t=1}^n$, we have that*

$$\mathbb{E}[L(\hat{\boldsymbol{\theta}}) - L(\tilde{\boldsymbol{\theta}})] \le C_1\left(\frac{1}{\sqrt{n}} + \sqrt{\mathbb{E}\left[\left(\pi(X) - \widehat{\pi}(X)\right)^2\right]\mathbb{E}\left[\sup_{y_0,a}\left(F_a(y_0|X) - \widehat{F}_a(y_0|X)\right)^2\right]}\right) \tag{11}$$

*where $C_1 = 4B\rho/c$.*

*Proof.* First note that

$$\mathbb{E}[\zeta(\boldsymbol{\theta}^{(t)}, Y_0^{(t)}, Z^{(t)})|\boldsymbol{\theta}^{(t)}] = \nabla\boldsymbol{\theta}^{(t)}L(\boldsymbol{\theta}^{(t)})$$

$$= \mathbb{E}\left[\nabla_{\boldsymbol{\theta}^{(t)}}g_{\boldsymbol{\theta}^{(t)}}(Y_0|X)\left(F_1\{g_{\boldsymbol{\theta}^{(t)}}(Y_0|X)|X\} - F_0(Y_0|X)\right)|\boldsymbol{\theta}^{(t)}\right].$$

We now aim to show that we are in the scenario of Lemma 5 with

$$\varepsilon^{(t)} = \hat{\zeta}(\boldsymbol{\theta}^{(t)}, Y_0^{(t)}, Z^{(t)}) - \mathbb{E}[\zeta(\boldsymbol{\theta}^{(t)}, Y_0, Z)|\boldsymbol{\theta}^{(t)}].$$

First we show that under Assumption 1(b), $L(\boldsymbol{\theta})$ is convex as a function of $\boldsymbol{\theta}$.

To this end we note that $\bar{\ell}(y_1, y_0, \boldsymbol{x})$ is convex w.r.t. $y_1$ as, by construction,

$$\partial_{y_1}\bar{\ell}(y_1, y_0, \boldsymbol{x}) = F_1(y_1|\boldsymbol{x}) - F_0(y_0|\boldsymbol{x})$$

which is increasing in $y_1$ for any $y_0, \boldsymbol{x}$. Furthermore for any $\boldsymbol{x}, y_0$ $g_{\boldsymbol{\theta}}$ is by construction affine in $\boldsymbol{\theta}$. Hence, as the composition of an affine function and a convex function is convex, we have that

$$\ell(\boldsymbol{\theta}, y_0, \boldsymbol{x}) = \bar{\ell}(g_{\boldsymbol{\theta}}(y_0|\boldsymbol{x}), y_0, \boldsymbol{x})$$

is convex w.r.t. $\boldsymbol{\theta}$. Hence as integrals of convex functions are convex, $L(\boldsymbol{\theta}) = \mathbb{E}[\ell(\boldsymbol{\theta}, Y_0, Z)]$ is also convex w.r.t. $\boldsymbol{\theta}$.

We also have that from Assumption 1(a) that $\bar{\zeta}(y_1, y_0, \boldsymbol{x}) \leq 1 + 1/c$ for all $y_1, y_0, \boldsymbol{x}$ combining this with Assumptions 1(b)&(c) we have that

$$\|v^{(t)}\| \leq \sup_{\boldsymbol{\theta}, y_0, \boldsymbol{z}} \|\boldsymbol{f}(y_0, \boldsymbol{x})\bar{\zeta}(\boldsymbol{\theta}^T \boldsymbol{f}(y_0|\boldsymbol{x}), y_0, \boldsymbol{z})$$

$$\leq \rho \cdot (1 + 1/c) \leq \frac{2\rho}{c}.$$

Meaning that we are in the setting of Lemma 5.

Taking expectations on over the result of the Lemma gives

$$\mathbb{E}[L(\hat{\boldsymbol{\theta}}) - L(\tilde{\boldsymbol{\theta}})] \leq \frac{2B\rho}{c\sqrt{n}} - \frac{1}{n} \sum_{t=1}^{n} \mathbb{E}\left[\left\langle \boldsymbol{\theta}^{(t)} - \tilde{\boldsymbol{\theta}}, \varepsilon^{(t)} \right\rangle\right]$$

and all we have remaining to do is bound $-\mathbb{E}\left[\left\langle \boldsymbol{\theta}^{(t)} - \tilde{\boldsymbol{\theta}}, \varepsilon^{(t)} \right\rangle\right]$. For this we have that

$$-\mathbb{E}\left\{\left\langle \boldsymbol{\theta}^{(t)} - \tilde{\boldsymbol{\theta}}, \varepsilon^{(t)} \right\rangle\right\} = -\mathbb{E}\left[\left\langle \boldsymbol{\theta}^{(t)} - \tilde{\boldsymbol{\theta}}, \mathbb{E}\left[\varepsilon^{(t)} \middle| \boldsymbol{\theta}^{(t)}\right] \right\rangle\right]$$

$$\leq \mathbb{E}\left\{\left\|\boldsymbol{\theta}^{(t)} - \tilde{\boldsymbol{\theta}}\right\| \left\|\mathbb{E}\left[\varepsilon^{(t)} \middle| \boldsymbol{\theta}^{(t)}\right]\right\|\right\} \quad \text{by the Cauchy-Schwartz inequality}$$

$$\leq \mathbb{E}\left\{\left\|\boldsymbol{\theta}^{(t)} - \tilde{\boldsymbol{\theta}}\right\| \mathbb{E}\left[\left\|\varepsilon^{(t)}\right\| \middle| \boldsymbol{\theta}^{(t)}\right]\right\} \quad \text{by Jensen's inequality}$$

$$\leq \frac{2\rho}{c} \sqrt{\mathbb{E}\left[\left|\pi(X) - \widehat{\pi}(X)\right|^2\right] \mathbb{E}\left[\sup_{\substack{y_0 \in \mathcal{Y}, \\ a \in \{0,1\}}} \left|F_a(y_0|X) - \widehat{F}_a(y_0|X)\right|^2\right]}$$

$$\cdot \mathbb{E}\left[\left\|\boldsymbol{\theta}^{(t)} - \tilde{\boldsymbol{\theta}}\right\|\right] \quad \text{by Lemma 6}$$

$$\leq \frac{4B\rho}{c} \sqrt{\mathbb{E}\left[\left|\pi(X) - \widehat{\pi}(X)\right|^2\right] \mathbb{E}\left[\sup_{\substack{y_0 \in \mathcal{Y}, \\ a \in \{0,1\}}} \left|F_a(y_0|X) - \widehat{F}_a(y_0|X)\right|^2\right]}.$$

with the final line coming from our projection step. Combining this with Lemma 5 gives our desired result. $\qquad \square$

### A.2.2 STRONGLY CONVEX CONVERGENCE

**Lemma 8.** *Let $L(\boldsymbol{\theta})$ be a strongly function w.r.t. $\boldsymbol{\theta}$ with strong convexity parameter $\eta$ and define $\tilde{\boldsymbol{\theta}} = \operatorname{argmin}_{\boldsymbol{\theta}} L(\boldsymbol{\theta})$. Assume that $\|\tilde{\boldsymbol{\theta}}\| \leq B$ for some $B > 0$. Define $\boldsymbol{\theta}^{(1)} = \boldsymbol{0}$ and inductively take*

$$\boldsymbol{\theta}^{(t+\frac{1}{2})} = \boldsymbol{\theta}^{(t)} - \mu_t v^{(t)} \qquad\qquad \boldsymbol{\theta}^{(t+1)} = \operatorname*{argmin}_{\boldsymbol{\theta}:\|\boldsymbol{\theta}\| \leq B} \|\boldsymbol{\theta} - \boldsymbol{\theta}^{(t+\frac{1}{2})}\|$$

*with $\mu_t = \frac{1}{\eta t}$ and $v^{(1)}, \ldots, v^{(n)}$ a sequence of RVs satisfying $\|v^{(t)}\| \leq \tilde{\rho}$ almost surely. Finally, take our parameter estimate to be $\hat{\boldsymbol{\theta}} = \frac{1}{n} \sum_{t=1}^{n} \boldsymbol{\theta}^{(t)}$.*

*Then we have that*

$$L(\hat{\boldsymbol{\theta}}) - L(\tilde{\boldsymbol{\theta}}) \leq \frac{\tilde{\rho}^2}{2\eta} \frac{1 + \log(n)}{n} - \frac{1}{n} \sum_{t=1}^{n} \left\langle \boldsymbol{\theta}^{(t)} - \boldsymbol{\theta}^*, \varepsilon^{(t)} \right\rangle$$

*where $\varepsilon^{(t)} := v^{(t)} - \nabla_{\boldsymbol{\theta}^{(t)}} L(\boldsymbol{\theta}^{(t)})$.*

*Proof of Theorem 5.* Define $\nabla^{(t)} := \nabla_{\boldsymbol{\theta}^{(t)}} L(\boldsymbol{\theta}^{(t)})$ so that $\mathbb{E}[v^{(t)}|\boldsymbol{\theta}^{(t)}] = \nabla^{(t)} + \varepsilon^{(t)}$ where $\nabla^{(t)}$ represents the unbiased gradient estimate and $\varepsilon^{(t)}$ represents the bias.

From Shalev-Shwartz and Ben-David (2014) section 14.4.1 we have that

$$\left\langle \boldsymbol{\theta}^{(t)} - \boldsymbol{\theta}^*, v^{(t)} \right\rangle \leq \frac{\mu_t}{2} \|v^{(t)}\|^2 + \frac{\|\boldsymbol{\theta}^{(t)} - \boldsymbol{\theta}^*\| - \|\boldsymbol{\theta}^{(t+1)} - \boldsymbol{\theta}^*\|^2}{2\mu_t}$$

Additionally we have

$$L(\hat{\boldsymbol{\theta}}) - L(\boldsymbol{\theta}^*) \leq \frac{1}{n} \sum_{t=1}^{n} L(\boldsymbol{\theta}^{(t)}) - L(\boldsymbol{\theta}^*) \quad \text{by Jensen's inequality.}$$

$$\leq \frac{1}{n} \sum_{t=1}^{n} \left\langle \boldsymbol{\theta}^{(t)} - \boldsymbol{\theta}^*, \nabla^{(t)} \right\rangle - \frac{\eta}{2} \|\boldsymbol{\theta}^{(t)} - \boldsymbol{\theta}^*\|^2 \quad \text{by strong convexity of } L$$

$$= \frac{1}{n} \sum_{t=1}^{n} \left\langle \boldsymbol{\theta}^{(t)} - \boldsymbol{\theta}^*, v^{(t)} \right\rangle - \frac{\eta}{2} \|\boldsymbol{\theta}^{(t)} - \boldsymbol{\theta}^*\|^2 - \frac{1}{n} \sum_{t=1}^{n} \left\langle \boldsymbol{\theta}^{(t)} - \boldsymbol{\theta}^*, \varepsilon^{(t)} \right\rangle$$

then from our prior result we get

$$= \frac{1}{n} \sum_{t=1}^{n} \frac{\mu_t}{2} \|v^{(t)}\|^2 + \frac{\|\boldsymbol{\theta}^{(t)} - \boldsymbol{\theta}^*\| - \|\boldsymbol{\theta}^{(t+1)} - \boldsymbol{\theta}^*\|^2}{2\mu_t} - \frac{\eta}{2} \|\boldsymbol{\theta}^{(t)} - \boldsymbol{\theta}^*\|^2 \boldsymbol{\theta}$$

$$- \frac{1}{n} \sum_{t=1}^{n} \left\langle \boldsymbol{\theta}^{(t)} - \boldsymbol{\theta}^*, \varepsilon^{(t)} \right\rangle$$

$$\leq \frac{1}{n} \sum_{t=1}^{n} \frac{\tilde{\rho}^2}{2\eta t} + \frac{1}{n} \frac{-\|\boldsymbol{\theta}^{(n+1)} - \boldsymbol{\theta}^*\|^2}{2\mu_n} - \frac{1}{n} \sum_{t=1}^{n} \left\langle \boldsymbol{\theta}^{(t)} - \boldsymbol{\theta}^*, \varepsilon^{(t)} \right\rangle$$

$$\leq \frac{\tilde{\rho}^2}{2\eta} \frac{1 + \log(n)}{n} - \frac{1}{n} \sum_{t=1}^{n} \left\langle \boldsymbol{\theta}^{(t)} - \boldsymbol{\theta}^*, \varepsilon^{(t)} \right\rangle.$$

$\square$

**Proposition 9.** *Suppose that assumption 1 holds and that $\|\boldsymbol{\theta}^*\| \leq B$ for some $B > 0$. Additionally now suppose that $p_1(y|\boldsymbol{x}) > \xi_2$ for all $y, \boldsymbol{x}$ and that the minimum eigenvalue of $\mathbb{E}\left[\boldsymbol{f}(Y_0, X)\boldsymbol{f}(Y_0, X)^\top\right]$ is greater than $\eta_2$.*

*Define $\boldsymbol{\theta}^{(1)} = \mathbf{0}$ and inductively take*

$$\boldsymbol{\theta}^{(t+\frac{1}{2})} = \boldsymbol{\theta}^{(t)} - \mu_t v^{(t)} \qquad\qquad \boldsymbol{\theta}^{(t+1)} = \underset{\boldsymbol{\theta}:\|\boldsymbol{\theta}\| \leq B}{\operatorname{argmin}} \|\boldsymbol{\theta} - \boldsymbol{\theta}^{(t+\frac{1}{2})}\|$$

*with $\mu_t = \frac{1}{\eta_2 \xi_2 n}$ and $v^{(t)} := \hat{\zeta}(\boldsymbol{\theta}^{(t)}, Y_0^{(t)}, Z^{(t)})$. Finally, take our parameter estimate to be $\hat{\boldsymbol{\theta}} = \frac{1}{n} \sum_{t=1}^{n} \boldsymbol{\theta}^{(t)}$. Then we have*

$$\mathbb{E}[L(\hat{\boldsymbol{\theta}}) - L(\tilde{\boldsymbol{\theta}})] \leq C_2 \left( \frac{1 + \log(n)}{n} + \sqrt{\mathbb{E}\left[\left(\pi(X) - \hat{\pi}(X)\right)^2\right] \mathbb{E}\left[ \sup_{\substack{y_0 \in \mathcal{Y} \\ a \in \{0,1\}}} \left(F_a(y_0|X) - \widehat{F}_a(y_0|X)\right)^2 \right]} \right)$$

*with $C_2 = \frac{\rho^2}{c\eta_2 \xi_2} + \frac{4B\rho}{c}$*

*Proof.* This is almost identical to the proof of Proposition 7. The only additional step is to prove strong convexity of $L(\boldsymbol{\theta})$.

We have that

$$
\begin{aligned}
\nabla_{\boldsymbol{\theta}} L(\boldsymbol{\theta}) &= \mathbb{E}\left[\nabla_{\boldsymbol{\theta}}(g_{\boldsymbol{\theta}}(y|\boldsymbol{x}))\left(F_1\left[g_{\boldsymbol{\theta}}(Y_0|\boldsymbol{x})|\boldsymbol{x}\right]\right) - F_0\left[y_0|\boldsymbol{x}\right]\right] \\
&= \mathbb{E}\left[\boldsymbol{f}(Y_0, X) \cdot \left(F_1\left[g_{\boldsymbol{\theta}}(Y_0|\boldsymbol{x})|\boldsymbol{x}\right]\right) - F_0\left[y_0|\boldsymbol{x}\right]\right] \\
\Rightarrow \nabla_{\boldsymbol{\theta}}^2 L(\boldsymbol{\theta}) &= \mathbb{E}\left[\boldsymbol{f}(Y_0, X)\boldsymbol{f}(Y_0, X)^\top \cdot \partial_{g_{\boldsymbol{\theta}}(Y_0|X)}\left(F_1\left[g_{\boldsymbol{\theta}}(Y_0|X)|X\right]\right) - F_0\left[Y_0|X\right]\right] \\
&= \mathbb{E}\left[\boldsymbol{f}(Y_0, X)\boldsymbol{f}(Y_0, X)^\top \cdot p_1\left\{g_{\boldsymbol{\theta}}(Y_0|X)\right\}\right] \\
&\geq \xi_2 \mathbb{E}\left[\boldsymbol{f}(Y_0, X)\boldsymbol{f}(Y_0, X)^\top\right]
\end{aligned}
$$

which by our assumptions has minimum Eigenvalue greater than $\xi_2 \eta_2$. Hence $L(\boldsymbol{\theta})$ is strongly convex with parameter $\eta := \xi_2 \eta_2$.

Now we can proceed as in Theorem 3 to obtain

$$
\mathbb{E}[L(\hat{\boldsymbol{\theta}}) - L(\tilde{\boldsymbol{\theta}})] \leq \frac{\rho}{c\eta_2\xi_2}\frac{1 + \log(n)}{n} - \mathbb{E}\left[\frac{1}{n}\sum_{t=1}^{n}\left\langle \boldsymbol{\theta}^{(t)} - \tilde{\boldsymbol{\theta}}, \varepsilon^{(t)}\right\rangle\right].
$$

Then using Lemma 6 and following an identical approach to Proposition 7 we get our result. □

### A.2.3 PROOF OF THEOREM 3

*Proof of Theorem 3.* This result is simply the concatenations of Propositions 7 & 9. □

### A.2.4 PROBABILITY BOUNDS

We first state a version of Azuma-Hoeffding bound which will be useful for our work. This Lemma is a slight modification of the version found in Wainwright (2019).

**Lemma 10** (Azuma-Hoeffding). *For $n \in \mathbb{N}$, let $W^{(1)}, \ldots, W^{(n)}$ be a Martingale difference sequence with respect to filtration $\{\mathcal{F}^{(t)}\}_{t=1}^{n}$*

*Suppose also that $\left|W^{(t)}\right| \leq \tilde{\rho}$ a.s. for all $t \in [n]$. We then have that for any $\delta > 0$*

$$
\mathbb{P}\left(\frac{1}{n}\sum_{t=1}^{n}W^{(t)} \leq \sqrt{\frac{2\log(1/\delta)}{n}}\right) \geq 1 - \delta.
$$

**Remark 7.** *For $W^{(t)}$ to be a martingale difference sequence we must have that $W^{(t)}$ is $\mathcal{F}^{(t)}$ measurable, $\mathbb{E}[|W^{(t)}|] < \infty$, and $\mathbb{E}[W^{(t)}|\mathcal{F}^{(t-1)}] = 0$ a.s. .*

We now get finite sample probability result in the setting of the first part of Theorem 3. For clarity we restate this setting in the result.

**Proposition 11.** *Suppose that assumption 1 holds and that $\|\tilde{\boldsymbol{\theta}}\| \leq B$ for some $B > 0$. For $t \in [n]$, define $\boldsymbol{\theta}^{(t)}$ inductively by*

$$
\boldsymbol{\theta}^{(t+\frac{1}{2})} = \boldsymbol{\theta}^{(t)} - \mu_t v^{(t)} \qquad\qquad \boldsymbol{\theta}^{(t+1)} = \underset{\boldsymbol{\theta}:\|\boldsymbol{\theta}\|\leq B}{\operatorname{argmin}}\|\boldsymbol{\theta} - \boldsymbol{\theta}^{(t+\frac{1}{2})}\|
$$

*with $\boldsymbol{\theta}^{(1)} = 0$, $\mu_t = \frac{Bc}{2\rho\sqrt{n}}$, and $v^{(t)} := \hat{\zeta}(\boldsymbol{\theta}^{(t)}, Y_0^{(t)}, Z^{(t)})$. Finally, define the parameter estimate as $\hat{\boldsymbol{\theta}} = \frac{1}{n}\sum_{t=1}^{n}\boldsymbol{\theta}^{(t)}$. Then if $\hat{\pi}, \widehat{F}_a$ are independent of $\left\{(Y_0^{(t)}, Z^{(t)})\right\}_{t=1}^{n}$, we have that for any $\delta > 0$, with probability at least $1 - \delta$,*

$$
L(\hat{\boldsymbol{\theta}}) - L(\tilde{\boldsymbol{\theta}}) \leq C_3 \frac{1 + \sqrt{\log(1/\delta)}}{\sqrt{n}} + \left|\pi(X) - \hat{\pi}(X)\right| \sup_{\substack{y_0 \in \mathcal{Y}, \\ a \in \{0,1\}}}\left|F_a(y_0|X) - \widehat{F}_a(y_0|X)\right|.
$$

*with $C_3 := 16\sqrt{2}\frac{B\rho}{c}$*

*Proof.* Again we are in the case of Lemma 5 with $\tilde{\rho} = 2\rho/c$ meaning we have that

$$L(\hat{\boldsymbol{\theta}}) - L(\tilde{\boldsymbol{\theta}}) \leq \frac{2B\rho}{c\sqrt{n}} - \frac{1}{n}\sum_{t=1}^{n}\left\langle \boldsymbol{\theta}^{(t)} - \tilde{\boldsymbol{\theta}}, \varepsilon^{(t)} \right\rangle \tag{12}$$

Now for $t \in [n]$ define the filtration $\{\mathcal{F}^{(t)}\}_{t=1}^{n}$ by $\mathcal{F}^{(t)} = \{\{\boldsymbol{\theta}^{(i)}\}_{i=1}^{t}, \widehat{\pi}, \widehat{F}_a\}$. Additionally define RVs $W^{(t)} = -\left\langle \boldsymbol{\theta}^{(t)} - \tilde{\boldsymbol{\theta}}, \varepsilon^{(t)} - \mathbb{E}[\varepsilon^{(t)}|\boldsymbol{\theta}^{(t)}, \widehat{\pi}, \widehat{F}_a] \right\rangle$.

Then we have that $\{W^{(t)}\}_{t=1}^{n}$ is a martingale difference process with respect to $\{\mathcal{F}^{(t)}\}_{t=1}^{n}$.

Furthermore we have that $\left\|v^{(t)}\right\| \leq \frac{2\rho}{c}$. Additionally $\left\|\nabla_{\boldsymbol{\theta}^{(t)}}L(\boldsymbol{\theta}^{(t)})\right\| \leq 2\rho$. Hence

$$\left\|\varepsilon^{(t)}\right\| \leq \frac{4\rho}{c}$$

$$\Rightarrow \left\|\varepsilon^{(t)} - \mathbb{E}\left[\varepsilon^{(t)}|\varepsilon^{(t)}|\boldsymbol{\theta}^{(t)}, \widehat{\pi}, \widehat{F}_a\right]\right\| \leq \frac{8\rho}{c}$$

$$\Rightarrow \left\|W^{(t)}\right\| \leq \frac{16B\rho}{c}.$$

As such we can apply the Azuma-Hoeffding inequality stated in Lemma 10 to get that

$$\mathbb{P}\left(\frac{1}{n}\sum_{t=1}^{n}W^{(t)} \leq C_3\sqrt{\frac{\log(1/\delta)}{n}}\right) \geq 1 - \delta.$$

with $C_3 = 16\sqrt{2}\frac{B\rho}{c}$. Furthermore we have that

$$\frac{1}{n}\sum_{t=1}^{n}W^{(t)} \leq C_3\sqrt{\frac{\log(1/\delta)}{n}}$$

$$\Rightarrow -\frac{1}{n}\sum_{t=1}^{n}\left\langle \boldsymbol{\theta}^{(t)} - \tilde{\boldsymbol{\theta}}, \varepsilon^{(t)} \right\rangle \leq C_3\sqrt{\frac{\log(1/\delta)}{n}} - \frac{1}{n}\sum_{i=1}^{n}\left\langle \boldsymbol{\theta}^{(t)} - \tilde{\boldsymbol{\theta}}, \mathbb{E}[\varepsilon^{(t)}|\boldsymbol{\theta}^{(t)}, \widehat{\pi}, \widehat{F}_a] \right\rangle$$

$$\Rightarrow -\frac{1}{n}\sum_{t=1}^{n}\left\langle \boldsymbol{\theta}^{(t)} - \tilde{\boldsymbol{\theta}}, \varepsilon^{(t)} \right\rangle \leq C_3\sqrt{\frac{\log(1/\delta)}{n}} + \frac{1}{n}\sum_{i=1}^{n}\left\|\boldsymbol{\theta}^{(t)} - \tilde{\boldsymbol{\theta}}\right\|\left\|\mathbb{E}[\varepsilon^{(t)}|\boldsymbol{\theta}^{(t)}, \widehat{\pi}, \widehat{F}_a]\right\|$$

by the Cauchy-Schwartz inequality. By Lemma 6 and the fact that $\left\|\boldsymbol{\theta}^{(t)} - \tilde{\boldsymbol{\theta}}\right\| \leq 2B$ this gives that

$$-\frac{1}{n}\sum_{t=1}^{n}\left\langle \boldsymbol{\theta}^{(t)} - \tilde{\boldsymbol{\theta}}, \varepsilon^{(t)} \right\rangle \leq C_3\left(\sqrt{\frac{\log(1/\delta)}{n}} + \left|\pi(X) - \widehat{\pi}(X)\right| \sup_{\substack{y_0 \in \mathcal{Y}, \\ a \in \{0,1\}}} \left|F_a(y_0|X) - \widehat{F}_a(y_0|X)\right|\right).$$

Hence by equation 12 we have that w.p. at least $1 - \delta$

$$L(\hat{\boldsymbol{\theta}}) - L(\tilde{\boldsymbol{\theta}}) \leq C_3\frac{1 + \sqrt{\log(1/\delta)}}{\sqrt{n}} + \left|\pi(X) - \widehat{\pi}(X)\right| \sup_{\substack{y_0 \in \mathcal{Y}, \\ a \in \{0,1\}}} \left|F_a(y_0|X) - \widehat{F}_a(y_0|X)\right|.$$

$\square$

# B   ADDITIONAL METHODS

## B.1   IPW APPROACH

Alternatively to our doubly-robust gradient estimator we can define an arguably simpler estimator which only depends on the propensity function $\pi$. This is done by defining

$$\zeta_{\mathrm{ipw}}(\boldsymbol{\theta}, y_0, \boldsymbol{z}) = \nabla_{\boldsymbol{\theta}}g_{\boldsymbol{\theta}}(y_0|\boldsymbol{x})\left(\frac{a}{\pi(\boldsymbol{x})}\mathbb{1}y \leq g_{\boldsymbol{\theta}}(y_0|\boldsymbol{x})\right) - \frac{1-a}{1-\pi(\boldsymbol{x})}\mathbb{1}y \leq y_0.$$

We then have that $\mathbb{E}[\zeta_{\text{ipw}}(\boldsymbol{\theta}, y_0, Z)|X = \boldsymbol{x}] = \nabla_{\boldsymbol{\theta}}\ell(\boldsymbol{\theta}, y_0, \boldsymbol{x})$. Meaning that Proposition 2 holds for $\zeta_{\text{ipw}}$ as well. From this we can define $\hat{\zeta}_{\text{ipw}}$ analogously to $\hat{\zeta}_{\text{dr}}$ and also use it in Algorithm 1. This is precisely the IPW procedure presented in our results.

In these results we see that the performance of this is very poor due to it's over reliance on inverse probability weighting which can be quite unstable.

### B.2 DIRECTLY EVALUATING THE LOSS

For validation purposes it can be useful to approximate the sample loss directly rather than its gradient. To obtain this from the gradient $\bar{\zeta}$ this we can split the objective into two parts, one involving all terms of $F_1(y_1|\boldsymbol{x})$ and all other terms.

As such we re-write $\bar{\zeta}$ as

$$\bar{\zeta}_{\text{dr}}(y_1, y_0, \boldsymbol{z}) := \underbrace{\frac{a}{\pi(\boldsymbol{x})}\left\{\mathbb{1}\{y \leq y_1\}\right\} - \frac{1-a}{1-\pi(\boldsymbol{x})}\left(\mathbb{1}\{y \leq y_0\} - F_0(y_0|\boldsymbol{x})\right) - F_0(y_0|\boldsymbol{x})}_{I_1}$$
$$+ \underbrace{\left(1 - \frac{a}{\pi(\boldsymbol{x})}\right)F_1(y_1|\boldsymbol{x})}_{I_2}$$

Now for the first term ($I_1$) we know that an anti-(weak)derivative is which keeps the loss continuous w.r.t. $y_1$ is

$$(y_1 - y)\left\{\frac{a}{\pi(\boldsymbol{x})}\left\{\mathbb{1}\{y \leq y_1\}\right\} - \frac{1-a}{1-\pi(\boldsymbol{x})}\left(\mathbb{1}\{y \leq y_0\} - F_0(y_0|\boldsymbol{x})\right) - F_0(y_0|\boldsymbol{x})\right\}$$

For the second term (which is continuous as a function of $y_1$) we can use the FTC to get an antiderivative of

$$\left(\frac{\pi(\boldsymbol{x}) - a}{\pi(\boldsymbol{x})}\right)\int_y^{y_1} F_1(t|\boldsymbol{x})\mathrm{d}t.$$

In fact we can also view the antiderivative of $I_1$ as the integral of $I_1$ between $y_1, y$.

Combining these we thus get

$$\bar{\ell}_{\text{dr}}(y_1, y_0, \boldsymbol{z}) = (y_1 - y)\left\{\frac{a}{\pi(\boldsymbol{x})}\left(\mathbb{1}\{y \leq y_1\}\right)\right.$$
$$\left. - \frac{1-a}{1-\pi(\boldsymbol{x})}\left(\mathbb{1}\{y \leq y_0\} - F_0(y_0|\boldsymbol{x})\right) - F_0(y_0|\boldsymbol{x})\right\}$$
$$+ \left(\frac{\pi(\boldsymbol{x}) - a}{\pi(\boldsymbol{x})}\right)\int_y^{y_1} F_1(t|\boldsymbol{x})\mathrm{d}t$$
$$\Rightarrow \mathbb{E}[\ell(\boldsymbol{\theta}, Y_0, Z)] = \mathbb{E}\left[(g_{\boldsymbol{\theta}}(Y_0|X) - Y)\left\{\frac{A}{\pi(X)}\left(\mathbb{1}\{Y \leq g_{\boldsymbol{\theta}}(Y_0|X)\}\right)\right.\right.$$
$$\left. - \frac{1-A}{1-\pi(X)}\left(\mathbb{1}\{Y \leq Y_0\} - F_0(Y_0|X)\right) - F_0(Y_0|X)\right\}$$
$$+ \left.\left(\frac{\pi(X) - A}{\pi(X)}\right)\int_y^{g(Y_0|X)} F_1(t|X)\mathrm{d}t\right]$$

We can then approximate the expectation via samples and the 1D integral via quadrature to get an approximation for the loss.

**Remark 8.** *The choice of $y$ for the lower bound of the integral is simply chosen to keep the size of the integral reasonable and to give the first term a simple form. Any choice of lower bound* not *depending upon $y_1$ would be valid.*

## C  ADDITIONAL DETAILS

### C.1  COMPLEXITY OF THE CQC VERSUS THE CCDF CONTRASTING FUNCTION

While not a strictly weaker notion, we do believe that a simple CQC function is a more natural notion than a simple CCDF contrasting function.

As a general case suppose we are in the potential outcomes framework so that $Y \equiv Y_A$ with $Y_0, Y_1$ representing our unobserved outcomes for an individual were the off or on treatment respectively. Suppose now that given $Y_0, X$ one can determine $Y_1$ as the following $Y_1 = f(Y_0, X)$ with $f$ an increasing function of $Y_0$ (a natural notion wherein those who perform better off treatment also perform better on treatment.) We then have that the CQC is given by $f$, in other words $g^*(y_0|\boldsymbol{x}) = f(y_0, \boldsymbol{x})$. Hence simplicity of $f$ translates directly to simplicity of the CQC.

Alternatively, for the CCDF contrasting function we get that

$$h(y_1, y_0, \boldsymbol{x}) = F_1(y_1|\boldsymbol{x}) - F_0(y_0|\boldsymbol{x})$$
$$= F_1(y_1|\boldsymbol{x}) - F_1(f(y_0, \boldsymbol{x})|\boldsymbol{x})$$

which does not necessarily cancel out to give a function of $f$ for all $y_0, y_1$. In fact the only case where we know this cancellation to occur is when $Y|X = \boldsymbol{x}, A = a$ are certain cases of uniform distributions.

### C.2  EXPERIMENTAL DETAILS

Here we provide additional details for our experiments. For our training we used 1,000 iterations of Adam with a learning-rate of 0.1 for any optimisation based approach. For estimation of the propensity score we used logistic regression with L2 regularisation.

For estimation of our CCDFs, we used kernel CCDF estimation. Specifically for a kernel $k : \mathcal{Y} \times \mathcal{X} \to \mathcal{X}$ and a sample $\left\{ \left( Y^{(i)}, X^{(i)} \right) \right\}_{i=1}^{n}$, we take

$$\widehat{F}_a(y|\boldsymbol{x}) := \frac{\sum_{i=1}^n k(\boldsymbol{x}, X^{(i)}) \mathbb{1}\{Y^{(i)} \leq y\}}{\sum_{i=1}^n k(\boldsymbol{x}, X^{(i)})}.$$

For our kernel we used an RBF kernel with bandwidth parameter chosen via grid-search testing on separate data against the true CCDF.

For hyper-parameter optimisation of our CQC model, with the linear and MLP models with our approach, the only hyperparameter that was tuned was the learning rate. This was set using an 80-20 splits for training and validation from half the data used in our training (the other half being used for nuisance parameter estimation.) As our validation loss we used the sample loss given in Appendix B.2. A choice was made to take the trimmed mean removing the top and bottom 5% of samples in order to avoid a small number of large samples dominating the loss. For the pre-existing inversion based method, the kernel bandwidth was chosen on validation data when comparing to the true CQC when the CQC was trained on balanced data so that no nuisance parameter estimates are required. While not possible in practical examples, this was done to ensure the inverting method was not hampered by poor hyperparameter selection.

Each experiment was ran on a single 4 core CPU with 16Gb of ram and took no longer than 240 minutes to run (less than 1-minute per iteration).

The employment scheme data used in Section 5 was originally provided in Autor and Houseman (2010) with a Creative Commons Attribution 4.0 International Public License found here: `https://www.openicpsr.org/openicpsr/project/113761/version/V1/view`.

The colon cancer data used in Appendix D.7 is provided as part of the R package survival and first introduced in Laurie et al. (1989) with no Licence provided.

## D  ADDITIONAL RESULTS

### D.1  1-DIM EXAMPLES

In this example our data set-up is as follows $X \sim N(0,1)$, $Y|X = x, A = 0 \sim N(\cos(6x), 1)$, $Y|X = x, A = 0 \sim N(2\cos(6x) + \gamma x, 4)$. Again in this case the marginal distributions contain

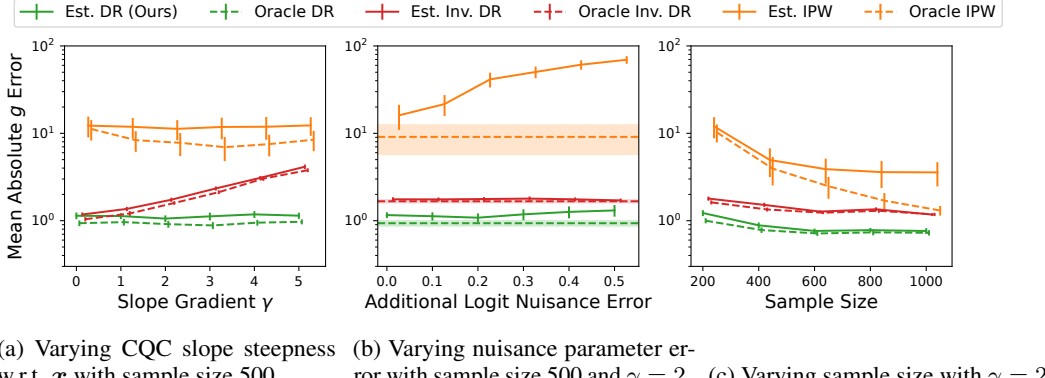

(a) Varying CQC slope steepness w.r.t. $\boldsymbol{x}$ with sample size 500.

(b) Varying nuisance parameter error with sample size 500 and $\gamma = 2$.

(c) Varying sample size with $\gamma = 2$

Figure 4: Truncated mean absolute error of CQC estimate for various methods with top and bottom 2.5% of runs removed alongside 95% C.I.s over 100 runs. Lower is best.

"complexity" via the high frequency sine term which persists into the CATE, CQTE, and CCDF contrasting function however the CQC is simple, being given by $g^*(y|\boldsymbol{x}) = 2y + \gamma x$. As in Section 4 we test estimation of this example with varying levels of $\gamma$ (representing steepness of our CQC), varying logit error on our nuisance parameter, and varying sample sizes. Due to a small number of outlier runs, for ease of interpretability, we present the truncated mean (where the largest and smallest 2.5% of results for each method removed) alongside 95% confidence intervals in figure 4. For transparency, we also present the standard mean with 95% confidence intervals in Figure 5.

Here we see identical patterns to our previous 10-dimensional example presented in Figure 2, with our approach (Est. DR) performing strongest in almost all cases. We again see that as the CQC gets steeper (Figure 4a) our estimation error stay relatively unchanged while the estimation error of the inverting approach gets worse.

As we vary nuisance parameter estimation error (Figure 4b), observe that Est. DR performs best at all levels. Despite this, we again observe that there is no discernible difference between Est. Inv. DR and Oracle Inv. DR whereas Est. DR does seem to perform marginally worse than Oracle DR. This does seem to support the hypothesis that Est. Inv. DR is more robust to nuisance parameter estimation error. We do still see evidence of robustness in Est. DR however as it is still minimally affected by nuisance parameter estimation error when compared to Est. IPW (which is not doubly robust.)

In Figure 4c, we see our approach, Est. DR, having the smallest Mean absolute error across all sample sizes.

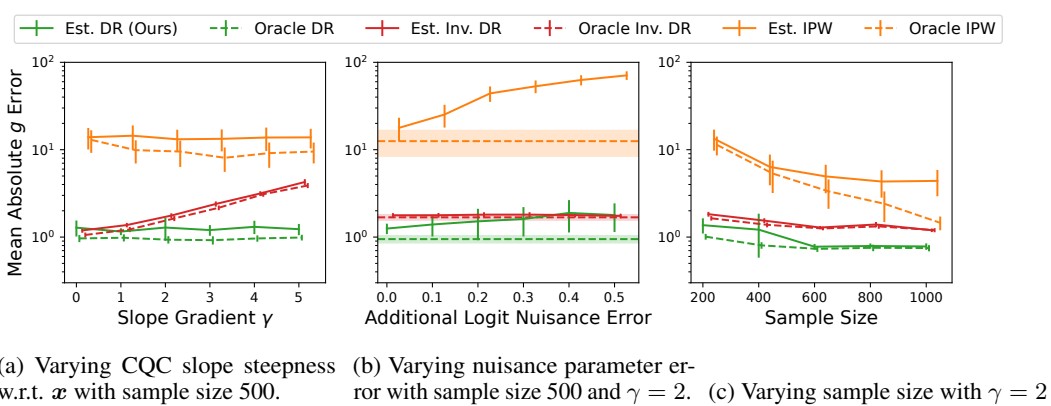

(a) Varying CQC slope steepness w.r.t. $\boldsymbol{x}$ with sample size 500.

(b) Varying nuisance parameter error with sample size 500 and $\gamma = 2$.

(c) Varying sample size with $\gamma = 2$

Figure 5: Mean absolute error of CQC estimate for various methods with 95% C.I.s over 100 runs. Lower is best.

## D.2  10-DIM EXPERIMENT

Here we present additional results from our 10 dimensional experiment introduced in Section 4.

### D.2.1  S-LEARNER AND CQTE APPROACH

Here we introduce additional comparators specifically in the form of an S-Learner and the CQTE estimator of Kallus and Oprescu (2023).

**S-Learner**  The S-Learner works by finding the value of $y_1$ which sets $\hat{h}(y_1, y_0, \boldsymbol{x}) = \widehat{F}_1(y_1|\boldsymbol{x}) - \widehat{F}_0(y_0|\boldsymbol{x}) = 0$. This can equivalent be thought of as taking our estimator to be $\widehat{F}_1^{-1}(\widehat{F}_0(y_0|\boldsymbol{x})|\boldsymbol{x})$ where $\widehat{F}_a^{-1}$ is computed by inverting $\widehat{F}_1$.

The results are presented in Figure 6. As we can see that Separate approach performs comparably to the DR approach in most settings except for the case when the slope parameter is set to 0. We can potentially understand this in terms of the derivative of our CQC w.r.t. $\boldsymbol{x}$. We have that $\nabla_{\boldsymbol{x}} g^*(y_0|\boldsymbol{x}) = \gamma\boldsymbol{v}$. Alternatively we see that $\nabla F_0(y_0|\boldsymbol{x}) = \nabla\Phi(y - \sin(\pi\boldsymbol{v}^\top\boldsymbol{x})) = f(y - \sin(\pi\boldsymbol{v}^\top\boldsymbol{x})) \cdot \pi\boldsymbol{v}$ where $\Phi, f$ are the CDF and density of a 0 means standard deviation 1 Gaussian. As such while the CQC is a simpler function, its derivative can be on a larger scale than that of the CQC making it more difficult to estimate from the perspective of Nadraya-Watson (NW) estimation. As such an approach which estimates the CQC using NW estimation (as the inverting approach does) will get minimal benefit over estimating the two CCDFs separately and using this as its estimate.

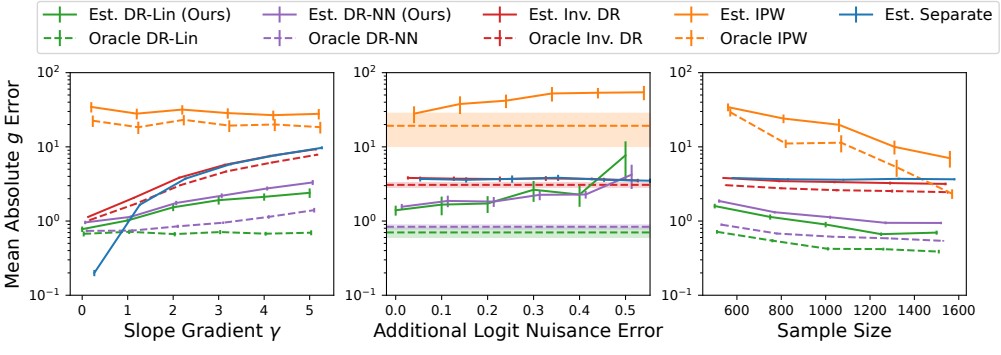

(a) Varying CQC slope steepness w.r.t. $\boldsymbol{x}$ with sample size 500.

(b) Varying nuisance parameter error with sample size 500 and $\gamma = 2$.

(c) Varying sample size with $\gamma = 2$

Figure 6: Mean absolute error of CQC estimate for various methods with 95% C.I.s over 100 runs.

**CQTE Estimator**  We also compare to the CQTE estimator of Kallus and Oprescu (2023). For estimation of each nuisance parameter and the final regression we use the same approach as used for the inverting estimator of Givens et al. (2024). The CQTE also requires estimation of the conditional density of $Y|X$ as the quantiles. That is for $a \in \{0, 1\}$, $p_{Y|X,A=a}(F_a^{-1}(\alpha|\boldsymbol{x}))$ for a given value of $\alpha$ our specified quantile level. To rule out poor performance due to poor estimation of this additional nuisance parameter we use its exact value for both the oracle and estimated approach. To compare this estimator to our CQC estimate we use the identity

$$g(y|\boldsymbol{x}) = \tau_q(F_0(y|\boldsymbol{x})) + F_0(y|\boldsymbol{x})$$

to transform the CQTE estimate using the exact CCDF. Additionally as the CQTE estimator is constructed to learn the CQTE for a specific quantile, for each run we fix the quantile that we will test the estimator on. We do not change the training procedure of the other estimators.

Results for this experiment are given in Figure 7 as we can see the CQTE approach performs comparably to the inverting approach and performs significantly worse than our direct estimator in almost all settings. Interestingly the Oracle and estimated approaches appear indistinguishable we could be due to using exact estimator of the conditional probability density function in both cases.

### D.3  VARYING HYPERPARAMETERS & COMPUTE TIME

### D.3.1  VARYING LEARNING RATE

Here we explore the effect of our choice of learning rate on our performance for our 10-dimensional experiment in Section 4. The results are presented in Figure 8.

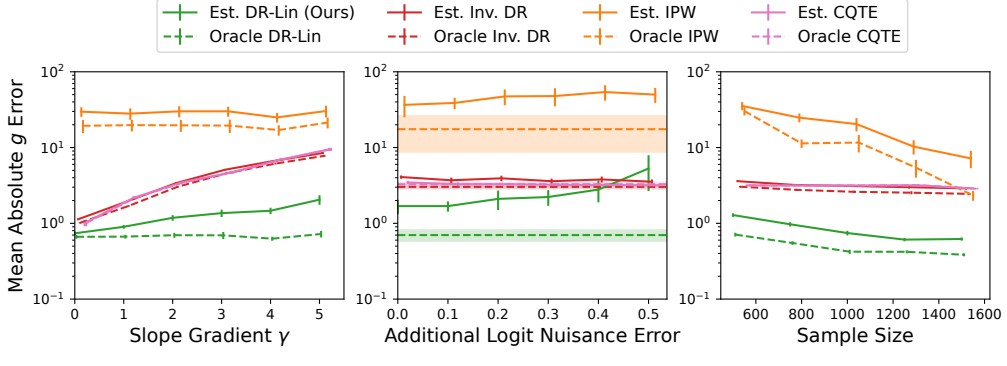

(a) Varying CQC slope steepness w.r.t. $x$ with sample size 500.

(b) Varying nuisance parameter error with sample size 500 and $\gamma = 2$.

(c) Varying sample size with $\gamma = 2$

Figure 7: Mean absolute error of CQC estimate for various methods with 95% C.I.s over 100 runs.

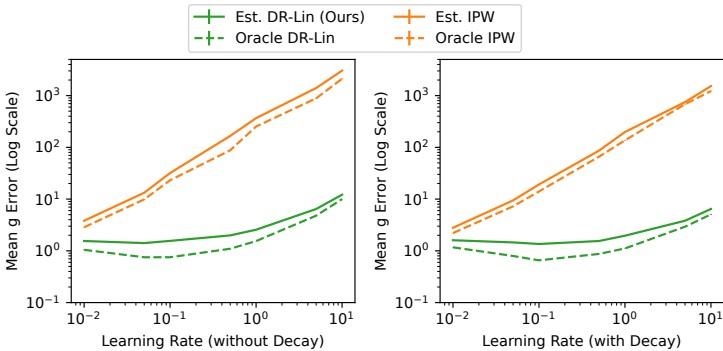

Figure 8: Mean absolute error of CQC estimate for various methods as learning rate increases. 95% C.I.s included. For the right figure a learning rate decay was also introduced

As we can see, for our DR method, higher learning rates can hamper performance although the method does not seem excessively sensitive to learning rates. By contrast the IPW approach gets drastically worse as learning rate increases. We also see that adding learning rate decay can further mitigate the effect of the learning rate on performance. For our main experiment we chose our learning rate via a validation procedure using the test loss discussed in Appendix B.2.

### D.3.2 VARYING ITERATION NUMBER

He we explore the rate at which our method converges. In Figure 9 we plot the convergence of our method for the IPW and DR approaches with oracle and estimated nuisance parameters.

We see that our DR approach converges within about 150 iterations while the IPW approach doesn't seem to converge at all or if it does converges to an incorrect value. We note that while 1000 iterations is very conservative, this still takes around 1 second with 1000 samples and so is reasonable to perform. In the following section we illustrate the time take for our new approach, demonstrating it to have more desirable dependence upon sample and test size.

### D.3.3 TIME TAKEN

In Figure 10 we plot the time take to train and evaluate various models for various number of training samples (left plot) and evaluation samples (right plot). We see that for small training and evaluation samples the previous inverting approach is quicker due to not having a distinct training sample however we can see that overall it has less desirable dependency on the training and evaluation samples, with the computational cost being $O(n^2 m)$ compared to $O(nT + m)$ for our approach with $n =$ sample size, $m =$ evaluation size, $T =$ iterations. Throughout we kept iterations fixed at an overly conservative 1000.

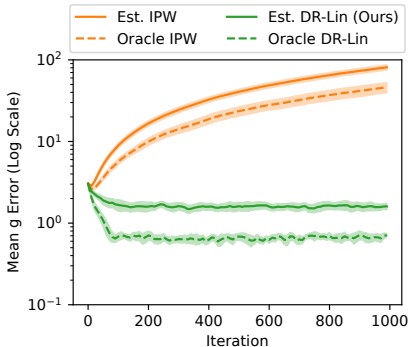

Figure 9: Mean absolute error of CQC estimate for various methods over iteration number. 95% C.I.s included.

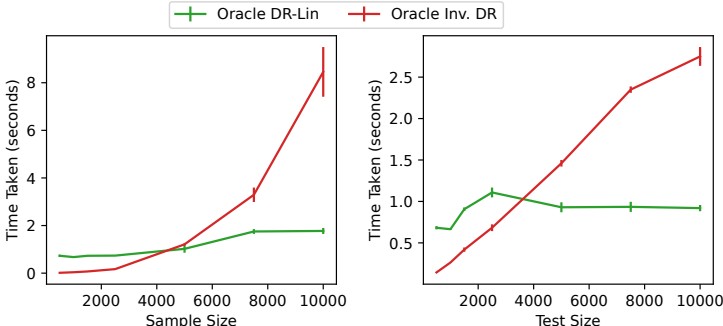

Figure 10: Mean time taken for training and evaluation of our gradient approach and the inverting approach for varying number of training and evaluation samples. 95% C.I.s included.

## D.4 NUISANCE PARAMETER DEPENDENCE

Here we explore the dependence of our approach on the accuracy of our estimates. Specifically we fix either the propensity or the CCDFs at their true values and estimate the other alongside various levels of additional error. These results are presented in Figure 11.

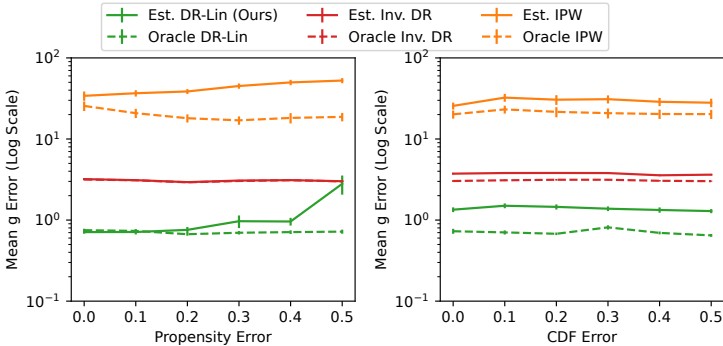

Figure 11: Mean absolute error of CQC estimate for various methods as nuisance error for Propensity and CCDF estimates increases separately. 95% C.I.s included.

We see that both CDF error and Propensity error have some effect on performance for our method. Interestingly with no additional error our propensity performs comparably to the oracle however additional propensity error can have a notable impact when it gets too large. Interestingly for the CCDFs, additional error doesn't seem to impact performance but our estimated approach performs

significantly worse than our oracle estimator suggesting that our estimate for the CCDFs is already quite poor. For the inverting approach, increased propensity error seems to have no effect while the effect of the estimated CCDF is small but statistically significant.

## D.5 $Y_0$ SAMPLING METHOD

Here we explore the impact of our sampling choice on $Y_0$ as discussed in Remark 6. Specifically we sample $Y_0$ in 3 different ways. Firstly we sample $Y_0$ uniformly from the range of the 5%-95% quantile of $Y_0$ and call this method "Uniform". Secondly we sample $Y_0$ uniformly with replacement from our $Y$ samples with $A = 0$ to approximately sample from $Y|A = 0$ and call this method "Unconditional". Finally we sample exactly from $Y|X = X^{(i)}, A = 0$ for each $X^{(i)}$ using the true inverse CDF and call this method "Conditional". Performance over various sample sizes are presented in Figure 12. As we can see the sample choice seems to have little impact on performance with the "Uniform" approach potentially performing marginally worse although this is not statistically significant for all sample sizes.

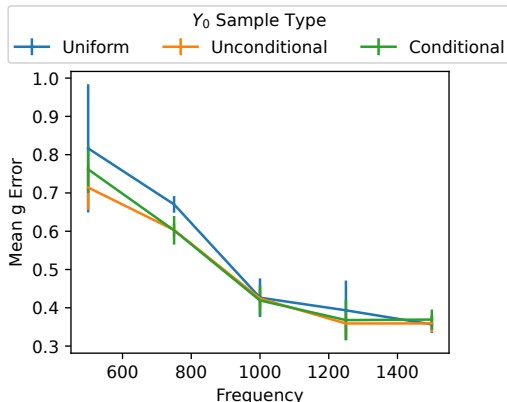

Figure 12: Mean absolute error of CQC estimate for various $Y_0$ sampling choices as sample size increases. 95% C.I.s included.

## D.6 EMPLOYMENT SCHEME EXAMPLE

Here we provide the parameters themselves for our aforementioned employment example.

Table 1: Table presenting the covariates from our CQC estimate plotted in Figure 3. The mode is $g_{\boldsymbol{\theta}}(y|x) = \theta_{\text{int, shift}} + \theta_{\text{age, shift}}x + (\theta_{\text{int, scale}} + \theta_{\text{age, scale}}x)\,y$

|  | Parameter Type | |
| --- | --- | --- |
| **Covariate** | Shift | Scale |
| Intercept | 1.43 | 1.74 |
| Age | 0.032 | -0.017 |

We can see the overall shape of the CQC represented in the parameters. Firstly we see that the scale term is significantly larger than 1 at the intercept and will continue to be larger than 1 for all values of age thus representing an increase in earning improvement as non-intervention earnings increase. We also see this increase in earning improvement decrease as a function of age as the age scale parameter is negative. We can easily see how one could generalise this to multiple covariates. For interpretability it perhaps makes sense to normalise both $y$ and $x$ for all parameters to be on a comparable scale and give the intercept a more natural interpretation.

## D.7 COLON CANCER EXAMPLE

We additionally apply our trial to data from a clinical trial on the the effect of colon cancer treatment on survival time/time to remission. This dataset was originally introduced in Laurie et al. (1989) and can be found in the "survival" package in R and loaded with the line `data(colon, package="survival")`. It was also previously studied via the CQC in Givens

et al. (2024). The dataset consists of 929 patients who are randomised to receive either treatment or control. The time until their death, recurrence of their cancer, or the end of the trial was then recorded alongside which one of these 3 outcomes occurred. The longest recorded time an individual participated in the trial was 3329 days. We take our response $(Y)$ to be the time until their event/end of the trial and a 1-dim covariate $(X)$ of the participants age upon trial entry.

As previous analysis of this trial showed the CQC to be distinctly nonlinear, here we fit the CQC using a fully connected Neural Network (NN). This NN takes in $y_0, \boldsymbol{x}$ as two separate features and then consists of two fully connected hidden layers of 20 nodes each and tanh activation functions. One again we estimate $g^*$ and then use this to estimate $\Delta(y|\boldsymbol{x}) = g^*(y|\boldsymbol{x}) - y$. The results of this estimation are given in Figure 13. For comparison we provide the estimated CQC via the existing inversion procedure in Figure 14

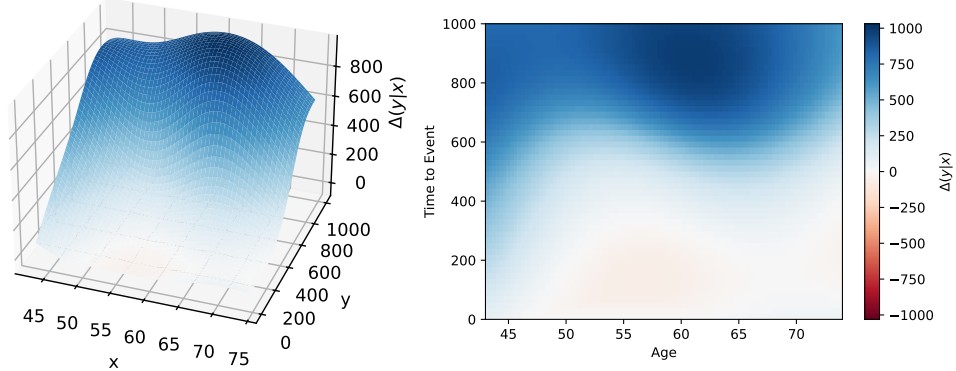

Figure 13: Surface plot and heat plot of $\Delta(y|\boldsymbol{x})$ over $y, \boldsymbol{x}$ for colon cancer trial data with $X =$Age, $Y =$Time to Event.

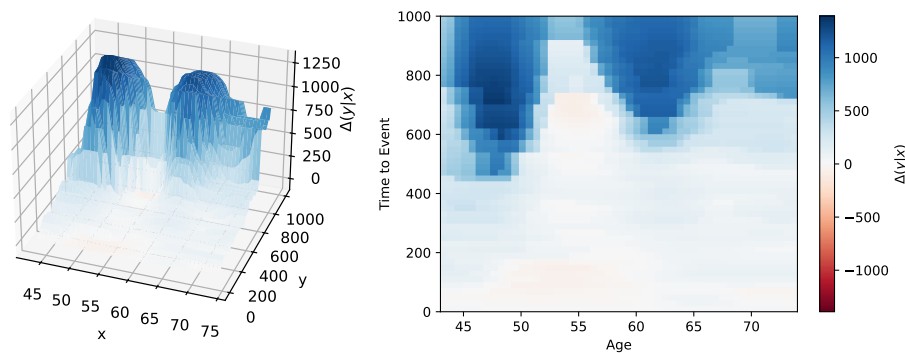

Figure 14: Surface plot and heat plot of $\Delta(y|\boldsymbol{x})$ over $y, \boldsymbol{x}$ for colon cancer trial data with $X =$Age, $Y =$Time to Event.

Here we see a very interesting pattern in which for the a reasonable range of the untreated response, the treated response is no difference and then there is a sudden increase in the treated response. This seems to suggest a relatively binary treatment outcome in which some people do not respond at all to treatment while others see a marked improvement. Interestingly, we also see that individuals younger than 50 seem to be most likely to see an improvement in their outcome while the strongest improvement seems to come for a smaller number of individuals between the ages of 56-66. This could partially be a result of the censoring as the largest values present on the graph are over 1,000 days larger than the untreated survival time of 1,000 days which, in total is reaching the longer end of follow-up. All of this aligns closely with the estimate CQC via the existing inversion approach presented in Figure 14 with the newer version providing a smoother and more readable estimate of the CQC.

## E    LLM USAGE

An LLM was used for minor editing of the papers prose. This was done solely for the purposes of conciseness and clarity.

