# OpenReview forum: "Direct Doubly Robust Estimation of Conditional Quantile Contrasts"
_ICLR.cc/2026/Conference — ICLR 2026 Poster_

### Official Review · Reviewer_YMsC · 2025-10-29

**Soundness:** 3
**Presentation:** 3
**Contribution:** 2
**Rating:** 6
**Confidence:** 3

**Summary:**

This paper provided an estimation framework for the conditional quantile comparator (CQC). The CQC estimand offers another dimension of heterogeneity compared to other HTE estimand (e.g. CATE) into it allows interpretation along the level of the untreated potential outcome. The proposed estimator reframe the Z-estimation problem into a loss minimization problem, and constructs a doubly robust pseudo-outcome for the gradient. The estimator is learned by taking a series of gradient update steps without explicitly evaluating the loss. The authors provided convergence analysis for the proposed estimator and demonstrated improved performance over existing baselines.

**Strengths:**

- Provided theoretical guarantee on the convergence rate, and show that the difference in loss is doubly robust in nuisance estimation error.
- Provided detailed overview/comparison with prior works.
- The authors discussed some key limitations in the paper.

**Weaknesses:**

- It is more sensitive to nuisance estimation error compared to the plug-in/invert approach as the doubly robust rate is proven for the loss.
- Since the only the gradient is calculated (the loss is never evaluated), it is hard to asses whether taking the gradient steps are sufficient.
- Would be stronger if the paper also included experiments on hyper-parameters like step size or number of gradient update steps.

**Questions:**

- $\tilde{\theta}$ in Theorem seems to be undefined, is it the optimal solution within the radius B?
- Is the proposed method sensitive to hyper-parameters (especially since there is clear evaluation to guide when to stop the gradient updates) like the step size?
- What was the main technical difficulty when proving convergence rate for the CQC it self?
- What causal quantity does the CQC correspond to? Is it $\mathbb{E}[Y(1)|X,Y(0)=y]$? What are the identifying assumptions?

---

> ### Author Response · Authors · 2025-11-19
>
> Thank you for your helpful comments. We have submitted an updated manuscript and will address them now.
>
> ## Weakness 1
> > It is more sensitive to nuisance estimation error compared to the plug-in/invert approach as the doubly robust rate is proven for the loss.
>
> We appreciate that this is a weakness and acknowledge it in our limitations section however we do note that even with the worse robustness it does seem to consistently outperform the inverting approach.
>
> ## Weakness 2
> > Since the only the gradient is calculated (the loss is never evaluated), it is hard to asses whether taking the gradient steps are sufficient.
>
> In appendix B.2. we propose a way to approximate the loss function via quadrature. We have now added a remark on line 263 to acknowledge (what?). We actually use this to compute a test loss in order to choose an optimal learning rate for our procedure. We intend to include an illustrative example of doing this within the appendix as well. Additionally for convergence purposes, we can use the norm of the gradient to assess convergence, especially in the convex case in the assumptions of our theoretical results.
>
> ## Weakness 3/ Question 2
> > Would be stronger if the paper also included experiments on hyper-parameters like step size or number of gradient update steps.
>
> Thank you for the question/comment. We have included an experiment illustrating the effect of learning rate/step size on performance in Appendix D.3.1 as we can see for our DR method the effect is relatively minimal but our IPW approach is quite strongly effected. In appendix B.2 we discuss a notion of test loss which can be used to optimise such hyperparameters.
>
> We also show convergence (where is it?) over number of gradient steps and see it to converge within 200 or so steps. In practice, with 1000 data points, running the method for 1000 iterations takes about 2 seconds and so we can comfortably be conservative in the number of test points required.
>
> ## Question 1
> Thank you for the question. $\tilde{\theta}$ is defined in definition 1 however we have now added reference to that in Theorem 1 as we a appreciate this is quite a while after it is introduced.
>
> ## Question 3
> >What was the main technical difficulty when proving convergence rate for the CQC it self?
>
> The main technical difficulty is translating error on our loss into error on the CQC, we do have various means of doing so as demonstrated in Proposition 1 and we briefly discuss the derived error for the CQC with density bounded below on line 350 however, given each assumption gives rise to a different bound on the CQC, we choose to leave our results as being generalised over the loss.
>
> ## Question 4
> > What causal quantity does the CQC correspond to?
>
> This is a good question. Fundamentally, as is the issue for the CQTE, we cannot frame our estimation as some causal quantity of the potential outcomes beyond $g^{*}(y|x)=F^{-1}_{Y(1)|X}(F(y|x)|x)$ (the second subscript is supposed to be $Y(0)\mid X$ but openreview is bugging out.) We note that if the treatment effect is given deterministically by some $Y(1)=\psi(Y(0)|X)$ with $\psi$ increasing in $Y(0)$, then $g=\psi$ i.e. the CQC can be literally interpreted as the effect of the treatment. However, due to quantiles requiring higher order information than the mean we cannot make direct inferences on the relationship between $Y(1)$ and $Y(0)$ without further assumptions.

---

### Official Review · Reviewer_A3DT · 2025-11-01

**Soundness:** 3
**Presentation:** 3
**Contribution:** 3
**Rating:** 6
**Confidence:** 3

**Summary:**

This paper proposes the first direct, doubly robust (DR) estimation method for the Conditional Quantile Comparator (CQC), an HTE estimand that bridges the gap between CATE and CQTE. The authors develop a novel loss function whose DR gradient can be estimated from data, allowing the CQC to be explicitly modeled and optimized, rather than being solved for indirectly via functional inversion.

**Strengths:**

1. This work provides a practical, direct, and more efficient alternative, making the CQC a much more usable tool.

2. The core method is novel. The idea of framing the CQC estimation problem as an M-estimation task by defining a loss $l$ such that $\partial_{y_1}l = h$ (with $h=F_1 - F_0$) is very interesting. Deriving the doubly-robust gradient $\zeta_{dr}$ (Proposition 2) provides a new class of estimators.

3. The method is theoretically solid. The paper provides finite sample bounds that formally demonstrate the estimator's double robustness.

4. The empirical study is thorough and shows that their direct method's error is low and constant, while the indirect method's error (which depends on $F_a$) is high and unstable.

**Weaknesses:**

1.  Practicality of model selection. The paper notes as a limitation that there is "no natural definition of test loss". The method optimizes based on an estimated gradient of the population loss, not a sample-based loss (like MSE). This makes standard validation and hyperparameter tuning very difficult. The paper suggests an approximation via quadrature (Appendix B.2), but this is complex and a significant practical barrier.

2. The algorithm (Algorithm 1) requires sampling test points $Y_0$ to compute the gradient. But there are some unclear points, as shown in the Questions 2&3.

**Questions:**

1. For Weakness 1, is it possible to derive any model evaluation metric, like those proposed in [1, 2], for evaluation? This is not the aim of this paper, but discussing this might bring some new insights for future research.

2. For Weakness 2, the loss $L(\theta)$ is an expectation over this $Y_0$ distribution. The paper suggests sampling from the control distribution ($Y|A=0$), but it's unclear how this choice affects the estimator's accuracy for $y_0$ values in the control group.

3. For Weakness 2, what is the exact sample algorithm? Is it random or dependent on some information? How the sampling algorithm would affect the final result?

Overall, I think this is a solid paper, and providing more explanations for the above questions might improve the quality.


[1] Unveiling the Potential of Robustness in Selecting Conditional Average Treatment Effect Estimators
[2] Empirical Analysis of Model Selection for Heterogeneous Causal Effect Estimation

---

> ### Author Response · Authors · 2025-11-19
>
> Thank you for your thoughtful comments. We have now uploaded a new revision of the paper and address each one specifically below.
>
> ## Weakness 1
> > Practicality of model selection. The paper notes as a limitation that there is "no natural definition of test loss".
>
> Thank you for the question. This was poorly phrased in our original manuscript as there is actually a notion of a test loss that we can use. In appendix B.2. we do discuss how 1-D quadrature can be used to obtain an approximation of the loss therefore allowing for some notion of a test loss. We do indeed use this test loss in order to choose the learning rate for our procedure as well (although we now show in Appendix D.3.1 that our main procedure is not too sensitive to learning rate choice.) We will highlight this loss estimation when our gradient estimator is introduced and change our framing in the limitations accordingly.
>
> ## Weakness 2
> > The algorithm (Algorithm 1) requires sampling test points  to compute the gradient.
>
> Thank you for the question. Our main algorithm used is to simply sample $Y_0$ from our existing $Y|A=0$ samples, in Appendix D.5 we provide an experiment where we try each of, sampling uniformly on the range of $Y$, sampling from our $Y|A=0$ samples and an idealised version where we sample exactly from $Y|A=0,X$. In these results we see that each of these perform very similarly to one another.
>
> Overall our ability to bound our loss via errors over $g^*$ is independent of this distribution choice. As we mention it can be seen as equivalent to choosing the distribution over $\alpha$ to minimise w.r.t. if you are estimating the CQTE over all $\alpha$.
>
> ## Questions
> We hope our above responses answer your questions but please let us know if you require further clarification.

---

### Official Review · Reviewer_2ZCv · 2025-11-01

**Soundness:** 2
**Presentation:** 3
**Contribution:** 2
**Rating:** 2
**Confidence:** 4

**Summary:**

This paper presents a novel approach for estimating the CQC, which removes the procedure of estimating an intermediate estimand followed by inversion, enabling the CQC estimate to be explicitly parametrized with enhanced interpretability. The authors cleverly transform the optimization scheme into a convex optimization problem by introducing
a loss function whose derivative with respect to y1 is the contrasting function. The upper bound of the loss is derived, and an estimator of the gradient is proposed. The authors illustrate that the estimated parameters obtained via gradient descent are doubly robust with respect to the loss, and demonstrate empirical results on simulated and real-world datasets.

**Strengths:**

1. The proposed framework enables parametrization of the CQC function,
providing a means to enforce structural assumptions on the model and
to represent the estimation error in terms of the complexity of the CQC
itself.
2. The idea of transforming the optimization scheme into a convex optimiza-
tion problem by introducing a loss function whose derivative with respect
to y1 is the contrasting function was fascinating.
3. Empirical results demonstrate improved performance.

**Weaknesses:**

1. The term ”direct CQC estimator” seems to be somewhat misleading, as the proposed estimator in Section 3.1 is defined with respect to the gradient rather than the estimand of interest, $g^∗$.

2. Also related to the point made in 1, and as the authors acknowledged in the limitations already, the doubly robustness proposed in Theorem 3 holds with respect to the loss function rather than the CQC estimate $g_\hat{\theta}$. As the estimand of interest is $g^∗$, it seems imperative to demonstrate the convergence rate of  $g_\hat{\theta}$. Also, the derivation of the doubly robust estimator appears fairly standard, given that the loss function is smooth; in fact, it closely mirrors existing results. Personally, I do not find much technical novelty in the manuscript, though I do acknowledge the conceptual value of the proposed framework itself.

3. The proposed method heavily depends on the previous work by Givens et al (2024), limiting its novelty and contribution.

**Questions:**

* Related to the second comment in \emph{Weaknesses}, does the $\sqrt{n}$ loss-consistency of $\hat{\theta}$ guarantee the consistency of $\hat{\theta}$ at sound rates? While the authors suggest that the result can be extended for a limited class of densities that are bounded below (lines 349--353), no guarantees are presented for a general class of probability densities.

 Minor typos:

- Line 117: $A = 0 \Rightarrow A = a$
- Line 142: provides an example of this by showing an example of this (redundant phrase)
- Line 201: $\mathcal{Y} \times \mathcal{X} \to \mathcal{X} \Rightarrow \mathcal{Y} \times \mathcal{X} \to \mathcal{Y}$
- Line 234: $g^*(y_0 | x) | x \Rightarrow g^*(y_0 | x)$
- Line 246: $g_\theta(g \mid x) \Rightarrow g_\theta(y_0 \mid x)$

---

> ### Author Response · Authors · 2025-11-21
>
> Thank you for your thoughtful comments.
>
> We have updated the manuscript to address some of them and will respond to each of them below.
>
> ## Weakness 1
> >The term ”direct CQC estimator” seems to be somewhat misleading,...
>
> We appreciate that the term direct here could be misleading. Here we take it to mean that our estimation procedure provides an estimate for the CQC which is not defined as some transformation of an intermediary estimand (as in the inverting case.) We note that for standard pseudo-outcome estimators such as that of Kennedy (2023) or Kallus et al. (2023). If we were to fit the primary estimator parametrically, this would be done by minimising least squares and so we would equally be using a gradient based scheme to fit the estimator.
> We have now added a section to our limitations acknowledging this different notion of directness on lines 476-481.
>
> ## Weakness 2
> >the doubly robustness proposed in Theorem 3 holds with respect to the loss function rather than the CQC estimate $g_\hat{\theta}$.
>
> We appreciate this comment and , as you noted, we pointed this out as a limitation of our work. In terms of converting our loss accuracy into accuracy on $g^*$ we note that, for bounded below densities we can do just that. Additionally for bounded above densities we get a similar bound in terms of the accuracy of our estimand on probability space as given in Proposition 1 which we believe to still be a valuable bound even being invariant to simple scaling of our estimand. While we agree that convergence on the error of the estimator would be valuable, we still see our convergence results as an important step towards understanding and constructing a direct estimator of the CQC.
>
> ## Weakenss 3
> > The proposed method heavily depends on the previous work by Givens et al (2024), limiting its novelty and contribution.
>
> We appreciate this comment and while our work does build upon the estimand introduced in the previous paper, the estimation method is completely distinct. We also believe that framing our problem via the gradient of an M-estimation problem is relatively unique among HTE estimation and therefore has value in and of itself helping to link quantile regression (which can be justified similarly) to doubly robust methods.
>
> ## Question 1
> > does the $\sqrt{n}$ loss-consistency of $\hat{\theta}$ guarantee the consistency of $\hat{\theta}$ at sound rates?
>
> Our result does guarantee consistency of $g^*$ at sound rates when the density is bounded below. We note that this is an assumption sometimes made within estimation problems such as in Gine et al. (2004) and Wainwright (2019; Section 14.4) We see more general convergence as an area for future work as we acknowledge in the limitations where we discuss the directness of our estimator.
>
> ## Typos
> Thank you for spotting these and they have all now been corrected in the manuscript.
>
> ## References
> Evarist Giné, Vladimir Koltchinskii, Joel Zinn "Weighted uniform consistency of kernel density estimators," The Annals of Probability, Ann. Probab. 32(3B), 2570-2605, (July 2004)
>
> Wainwright, M. J. (2019). Localization and uniform laws. In High-Dimensional Statistics: A Non-Asymptotic Viewpoint (pp. 453–484). chapter, Cambridge: Cambridge University Press.

---

### Official Review · Reviewer_MqaP · 2025-11-12

**Soundness:** 3
**Presentation:** 3
**Contribution:** 3
**Rating:** 6
**Confidence:** 2

**Summary:**

This paper introduces a new direct doubly robust estimator for the Conditional Quantile Comparator (CQC). Unlike prior approaches that estimate the CQC indirectly by inverting a contrast of conditional cumulative distribution functions, the authors present a direct parameterization and gradient-based estimation procedure that preserves double robustness while improving interpretability and computational efficiency.
Theoretical results include finite-sample bounds, convergence guarantees, and robustness proofs. Empirical validation shows improved estimation accuracy across simulated and real-world datasets (e.g., employment outcomes). The method also generalizes to both linear and neural network parameterizations.

**Strengths:**

The paper presents the first direct doubly robust estimator for the Conditional Quantile Contrast (CQC), offering a method that effectively connects theoretical causal inference principles with practical implementation. It contributes new finite-sample and convergence bounds, extending robustness theory in heterogeneous treatment effect estimation. The proposed algorithm is clearly articulated through a gradient-based optimization procedure (Algorithm 1), making it both conceptually transparent and computationally feasible. The authors validate their approach with comprehensive experiments that vary sample size, noise level, and functional complexity, and further demonstrate interpretability through a real-world employment dataset. Overall, the work is grounded in a solid mathematical foundation and builds meaningfully on existing causal inference literature, particularly the frameworks of Kennedy (2023) and Kallus (2023).

**Weaknesses:**

1. The real-world data analysis, while effectively illustrating the interpretability of the proposed estimator, lacks quantitative comparisons to other causal inference methods. The employment dataset experiment focuses on qualitative visualization of treatment heterogeneity but does not benchmark performance against either inversion-based CQC estimators or widely used CATE-based models such as TARNet, DragonNet, or BART. Including such comparisons would provide essential empirical context, clarifying whether modeling full conditional quantile contrasts yields measurable advantages over standard mean-based causal estimators in practical applications.

2. The introduction and abstract clearly present the statistical motivation behind estimating the Conditional Quantile Contrast (CQC) but do not effectively convey its practical significance. They could better highlight how CQC interpretation informs real-world decisions—such as in policy analysis, where understanding which subgroups benefit most or least from interventions is critical. Without a clear link to applied impact, the estimator’s broader relevance to practitioners and policymakers remains underemphasized.

3. There are some grammar errors:
Line 214: “AppendixA.1” → “Appendix A.1”
Line 331-332: “Proposition 1b)” → “Proposition 1(b)”
Line 361-362: “which their estimated equivalents” → “that uses their estimated equivalents”
Line 398: “effected” → “affected”
Line 409-410: “perform well” → “performs well”

**Questions:**

1. Could the authors expand the real-world analysis by including quantitative comparisons to other estimators? Specifically, how does the proposed method perform relative to both inversion-based CQC estimators and standard CATE-based models such as TARNet, DragonNet, or BART? Even though these approaches estimate different quantities, wouldn’t such comparisons help clarify whether modeling conditional quantile contrasts offers practical advantages over mean-based causal estimators? If you have reasons for not comparing those, could you explain reasons?

2. I think ablation studies isolating the contributions of each model component (e.g., nuisance models, parameterization choices) would strengthen the empirical claims - are there plans to include these analyses, or are there some reasons that this kind of study is not considered?

3. Are there some possible ways to formally quantify or evaluate interpretability beyond qualitative visualization?

4. Could the authors provide computational complexity comparisons (runtime, memory) versus the inversion method?

---

> ### Author Response · Authors · 2025-11-19
>
> Thank you for your comments, we have now updated the submission with some additional content and corrects per your recommendations and will go through each of these below.
>
> # Weakness 1
> > The real-world data analysis, while effectively illustrating the interpretability of the proposed estimator, lacks quantitative comparisons to other causal inference methods.
>
> We appreciate that empirical tests on real world data would be desirable. Unfortunately the nature of the estimand means that there is no natural notion of a test loss which we could use to test our methods. Regarding the various CATE estimation models, we are unable to perform 1 to 1 comparisons as they target entirely different objects meaning there is not a translateable notion of accuracy. For the CQTE, we have now added experiments comparing against the CQTE estimation procedure of Kallus et al. (2023) in Section D.2.1. We see that our direct estimator consistently our performs it with the CQTE approach being comparable to the inverting approach of Givens et al (2024).
>
> ## Weakness 2
> > The introduction and abstract clearly present the statistical motivation behind estimating the Conditional Quantile Contrast (CQC) but do not effectively convey its practical significance.
>
> Thank you for this comment. We agree that real world decision making is important and have added more discussion of its real world use into the introduction on lines 50-62.
>
> ## Weakness 3
> Typos - Thank you for finding these they have now been corrected.
>
> ## Question 1
> >Could the authors expand the real-world analysis by including quantitative comparisons to other estimators?
>
> See response to Weakness 1
>
> ## Question 2
> >I think ablation studies isolating the contributions of each model component (e.g., nuisance models, parameterization choices) would strengthen the empirical claims
>
> We appreciate this would be good to add. We have added results separately for estimating the propensity and the CCDF nuisance parameter in Appendix D.4. We can see that both nuisance parameters have an effect with the original estimate of the CDF and the additional error on the propensity both impacting performance but additional error on the CDF and our original estimate of the propensity not affecting performance. In terms of interpreting what benefit is the choice of parameterisation this is more difficult to do between the inverting and our method. We cannot directly apply the same non-parametric approach to our method due to it requiring gradients, conversely as the inverting approach doesn't estimate the CQC directly, we cannot use the direct parameterisation to parameterise it. We do intend to run an experiment with a parameterise CCDF contrasting function for the inverting approach however this will take new implementation and so we do not yet have results for this. We also note that the comparison between our Neural Network and Linear model CQC also explores this question and shows the Neural Network implementation to perform comparably.
>
> ## Question 3
> > Are there some possible ways to formally quantify or evaluate interpretability
>
> This is a good question and definitely something worth exploration. Fundamentally we view interpretability as a somewhat subjective metric. We do however highlight that our method has a superset of the interpretation techniques of the inverting approach as with our approach we can interpret by visualising over various values or by examining the parameter values whereas the inverting approach can only interpret via visualising over various values.
>
> ## Question 4
> > Could the authors provide computational complexity comparisons
> Thank you for the suggestion we have now added this to the Appendix section D.3.3. At a high level our approach is $O(nT+m)$ with $n=$training sample size, $m=$evaluation sample size, $T=$ max iteration number while the inverting approach is $O(n^2m)$. This is supported by our experimental results in Appendix D.3.3.

---

### Meta-Review · Area_Chair_YacP · 2026-01-12

**Summary:**

This paper proposes the first direct doubly robust estimator for the Conditional Quantile Comparator (CQC), addressing a key limitation of existing approaches that rely on estimating and inverting intermediate distributional quantities. By framing CQC estimation as an M-estimation problem with a carefully constructed loss whose gradient admits a doubly robust estimator, the authors enable explicit parameterization of the CQC, improved interpretability, and estimation error that depends on the complexity of the estimand itself rather than that of an upstream nuisance function. Theoretical results establish finite-sample convergence guarantees and double robustness properties, and the empirical evaluation demonstrates consistent improvements over inversion-based methods across a range of simulated settings, nuisance estimation regimes, and sample sizes, as well as meaningful insights in a real-world employment application.

The reviews are generally positive, highlighting the novelty of the direct estimation framework, the soundness of the theory, and the strength of the empirical results. One reviewer expressed skepticism and assigned a low score, primarily questioning the degree of technical novelty and noting that the doubly robust guarantee is stated with respect to the loss rather than directly for the CQC estimand. These concerns are thoughtfully addressed in the rebuttal: the authors clarify the notion of directness, explicitly acknowledge the limitations, and provide conditions under which consistency and convergence of the CQC itself can be derived. While this reviewer remains unconvinced, the disagreement reflects a conservative interpretation of novelty rather than unresolved issues of correctness or significance. Overall, I find that the paper makes a clear and meaningful contribution to heterogeneous treatment effect estimation and causal quantile analysis, and I recommend acceptance.

**Reviewer Concerns:**

everal reviewer concerns were addressed by the rebuttal. In particular, the authors clarified the notion of “direct” estimation, explaining how their approach avoids inversion of intermediate estimands and explicitly acknowledging alternative interpretations. They also responded to concerns about the doubly robust guarantee applying to the loss rather than directly to the CQC by making this limitation explicit and by providing conditions under which consistency and convergence of the CQC itself can be obtained. Presentation issues, including terminology, minor technical ambiguities, and typos, were corrected. In addition, the authors expanded the empirical section with further analyses and complexity comparisons, addressing questions about practical performance and implementation.

Some concerns remain partially outstanding. Reviewer 2ZCv’s skepticism regarding the degree of technical novelty relative to prior work, and the desire for stronger guarantees stated directly in terms of the CQC without additional assumptions, were not fully resolved and reflect a more conservative standard for novelty and theoretical completeness. However, these remaining issues concern emphasis and expectations rather than correctness, soundness, or empirical validity of the proposed method.

**Reviewer Scores:**

Reviewer MqaP.
Reviewer MqaP was generally positive and focused on empirical validation, presentation, and practical relevance. These concerns were largely addressed in the rebuttal through added experiments, expanded discussion of real-world impact, and corrections. I expect Reviewer MqaP would have maintained their score of 6 or increased it to 8 after full discussion.

Reviewer 2ZCv.
Reviewer 2ZCv expressed strong skepticism regarding novelty and the scope of the doubly robust guarantee and remained unconvinced after the rebuttal. While the authors clarified the notion of direct estimation and explicitly acknowledged the limitations of loss-based robustness, this reviewer’s core concerns reflect a conservative interpretation of novelty rather than unresolved technical errors. I expect Reviewer 2ZCv would have maintained their score of 2, or at most increased it to 4.

Reviewer A3DT.
Reviewer A3DT was positive about the conceptual contribution, theoretical soundness, and empirical performance, with concerns mainly about practicality and validation. These points were addressed through clarification and additional discussion. I expect Reviewer A3DT would have increased their score from 6 to 8 after discussion.

Reviewer YMsC.
Reviewer YMsC viewed the paper as technically sound and relevant but raised moderate concerns about practicality, validation, and positioning. The rebuttal addressed these points through clarification and additional discussion, but some reservations would likely remain. I therefore expect Reviewer YMsC would have maintained their score of 6 after full discussion, rather than making a substantial upward or downward change.

Overall, full discussion would likely have resulted in modest upward movement for some reviewers, with one remaining dissenting opinion and one reviewer remaining cautiously positive.

---

### Decision · Program_Chairs · 2026-01-26

Accept (Poster)